# Firing feature-driven neural circuits with scalable memristive neurons for robotic obstacle avoidance

Yue Yang[1,2,5], Fangduo Zhu[1,5], Xumeng Zhang [1] ✉, Pei Chen[1], Yongzhou Wang[2], Jiaxue Zhu[2], Yanting Ding[1], Lingli Cheng[1,2], Chao Li[1,2], Hao Jiang[1], Zhongrui Wang [3], Peng Lin [4], Tuo Shi[2], Ming Wang [1], Qi Liu [1,2] ✉, Ningsheng Xu[1] & Ming Liu[1,2]

Neural circuits with specific structures and diverse neuronal firing features are the foundation for supporting intelligent tasks in biology and are regarded as the driver for catalyzing next-generation artificial intelligence. Emulating neural circuits in hardware underpins engineering highly efficient neuromorphic chips, however, implementing a firing features-driven functional neural circuit is still an open question. In this work, inspired by avoidance neural circuits of crickets, we construct a spiking feature-driven sensorimotor control neural circuit consisting of three memristive Hodgkin-Huxley neurons. The ascending neurons exhibit mixed tonic spiking and bursting features, which are used for encoding sensing input. Additionally, we innovatively introduce a selective communication scheme in biology to decode mixed firing features using two descending neurons. We proceed to integrate such a neural circuit with a robot for avoidance control and achieve lower latency than conventional platforms. These results provide a foundation for implementing real brain-like systems driven by firing features with memristive neurons and put constructing high-order intelligent machines on the agenda.

Endowing robots flexibly interact with changing and unpredictable environments toward the completion of specific tasks is one of the goals in embodied intelligence[1,2]. The progress of deep learning algorithms and computing units promotes the field towards such an aspiration[3–7]. However, current intelligent robots remain incapable of performing as well as humans or animals, even in some reflex-like behaviors, such as object grasping or emergency escape. This occurs because these capabilities of biology are embedded in some functional neural circuits that have evolved over hundreds of millions of years. These neural circuits usually comprise neurons with disparate properties and functions connected in a variety of forms in the brain and perform specific tasks in an extremely efficient manner[8–11]. Inspired by such operations, several innovative strategies have been proposed to solve intelligent tasks in a more efficient and explicable way than conventional methods. For example, Lechner et al. proposed a novel algorithm with 19 control neurons, inspired by the topological structure of neural circuits of *Caenorhabditis elegans*, for autonomous vehicles[12]. By mapping the high-dimensional information from the camera into steering commands, the system achieves superior generalizability, interpretability and robustness with less source than conventional algorithms; Moro et al. demonstrated an object location neuromorphic circuit referring to the auditory localization neural

[1]State Key Laboratory of Integrated Chips and Systems, Frontier Institute of Chip and System, Fudan University, Shanghai 200433, China. [2]Key Laboratory of Microelectronics Device & Integrated Technology, Institute of Microelectronics of Chinese Academy of Sciences, Beijing 100029, China. [3]Department of Electrical and Electronic Engineering, The University of Hong Kong, Hong Kong 999077, China. [4]College of Computer Science and Technology, Zhejiang University, Zhejiang 310027, China. [5]These authors contributed equally: Yue Yang, Fangduo Zhu. ✉e-mail: xumengzhang@fudan.edu.cn; qi_liu@fudan.edu.cn

circuits in owl hunting, achieving the target location with extremely low power consumption[13]. These works illustrate the merits of emulating some neural principles in neural circuits but ignore the computational functional variety of neurons' firing features.

The cricket $AN_2$ neuron, a type of auditory interneuron, can detect ultrasonic signals and activate motor neurons to drive crickets to reflexively escape from predators (i.e., bats) by high-frequency firing feature (i.e., tonic bursting)[14]. Benefiting from the computational function of neurons' firing behaviors, crickets can complete the escape instantly, which provides a new idea for robots to accomplish tasks such as obstacle avoidance or even higher-order intelligent behaviors. Nevertheless, it is expensive to implement such a computation principle in conventional hardware platforms by simulating high-fidelity neurons, especially those with diverse firing behaviors that need rich dynamics[15–18]. Locally active memristors (LAMs), with rich dynamics[19–21], low power consumption[22] and good scalability[23–25], show intriguing potential to build Hodgkin-Huxley (H-H) neurons with excellent bio-plausibility[26–29]. However, current works typically focus on the emulation of neurons' firing behaviors, and the exhibition of firing features' computational capability remains an open question (see Supplementary Table 1). Thus, it is urgent to incorporate the firing features' computation policies into neural circuit architectures to leverage revolutionary strategies in embodied robot applications.

Here, inspired by the tonic bursting feature-driven (for convenience, spiking and bursting are used to refer to tonic spiking and tonic bursting, respectively, in the following) neural circuits involved in cricket avoidance behavior, we present a sensorimotor control neural circuit (SCNC) model including a biological selective communication scheme for robot obstacle avoidance. The SCNC comprises three types of neurons with a feedforward divergent connection structure: an ascending neuron that fires in mixed bursting and spiking features, a bursting-detection neuron (BDN) detecting bursting features, and a spiking-detection neuron (SDN) reacting to spiking features. To demonstrate such a model, we first construct an H-H neuron circuit underlying $NbO_2$ LAMs to serve as the ascending neuron. By

specifically designing the circuit parameters, we implement an input intensity-controlled transition between spiking and bursting features, in addition to the reported 23 natural firing behaviors. Moreover, attributed to the inherent stochastic switching in $NbO_2$ LAMs, the neuron features a probabilistic transition from spiking to bursting with increasing input intensity, exhibiting a mixed pattern. This characteristic affords the neuron a firing pattern-related encoding capability. Then, we introduce another two $NbO_2$ H-H neurons with specific parameters to build a complete SCNC and adopt the selective communication scheme that decodes the mixed firing patterns from the ascending neuron. To illustrate the feasibility of the SCNC for practical applications, we integrate it into a robot and test the resulting obstacle avoidance capabilities. Compared to conventional platforms, our SCNC achieves a reduction of more than one order of magnitude in latency. These results reveal the high efficiency of conducting tasks by emulating neural circuits and pave the way for building next-generation intelligent machines through biocomputing principles.

## Results

### Biological background

Invertebrates often rely on a simple neural circuit consisting of several neurons to control their activities[14]. In these circuits, sensory interneurons are considered to be detectors of predator signals, with functions that activate the descending neuron and dominate the avoidance behavior of organisms to the approaching danger[30–33]. Figure 1A shows the neural circuits of crickets that conduct such avoidance behavior, which consists of auditory receptors, auditory interneurons, and descending motor neurons. The auditory receptors receive ultrasonic information at cricket ears located on the tibiae of the forelegs and generate action potentials[34,35]. Then, the generated action potentials are transmitted to the auditory interneurons (named $AN_2$) located at the prothoracic ganglion and are encoded as spiking features to transmit to the descending neurons[36,37]. Next, two descending neurons decode the received signals and generate specific spiking features to control the avoidance behavior of the crickets[38,39].

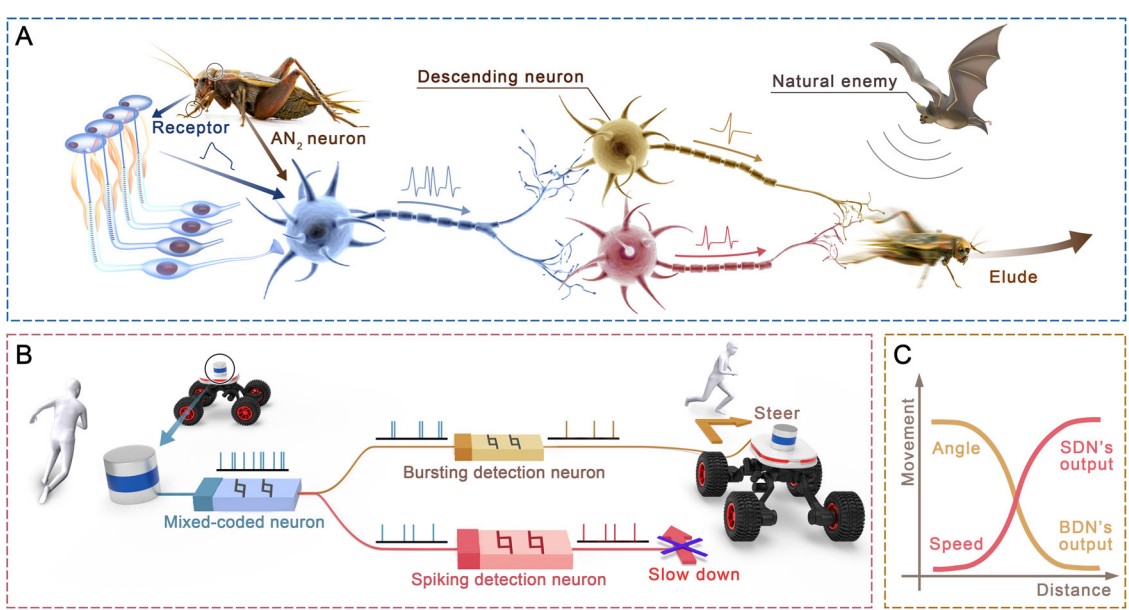

**Fig. 1 | Neuroinspired SCNC for robot obstacle avoidance. A** Schematic of the biological neural circuit associated with escape behavior in crickets. Ultrasonic signals are received by the receptors, transmitted to $AN_2$ neurons for encoding and then decoded by the descending neurons to control the behavior of crickets. The high-frequency "bursting" feature of $AN_2$ neurons causes crickets to escape. **B** Schematic of the neuro-inspired SCNC. The distance information from the obstacle is transmitted to the ascending memristive H-H neuron (mixed-coded neuron, MCN) through LiDAR, encoded as spiking features, and then decoded by two descending memristive H-H neurons (bursting detection neuron, BDN and spiking detection neuron, SDN) to control the steering and actuating of the robot, respectively. **C** Obstacle avoidance scheme related to the robot-obstacle distance. The steering angle and moving speed are controlled by the firing frequencies of BDN and SDN, respectively.

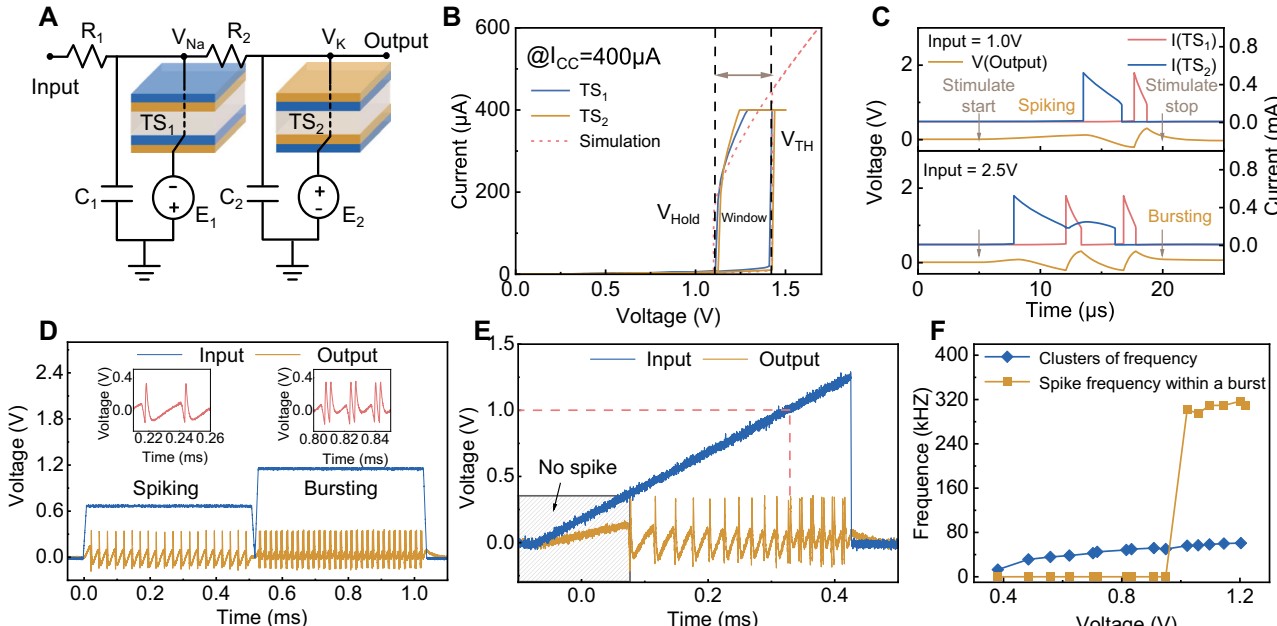

**Fig. 2 | H-H neuron circuit based on NbO2 memristors and the firing behaviors.**
**A** The H-H neuron consists of two resistors ($R_1$, $R_2$), two capacitors ($C_1$, $C_2$), and two
TS devices ($TS_1$, $TS_2$) with opposite bias voltages provided by two voltage sources
($E_1$, $E_2$). **B** Typical volatile threshold switching behavior of two TS devices used in
the H-H neurons. Both TS devices exhibit similar $V_{TH}$ and $V_{Hold}$. **C** Top panel: Schematic
of TS devices' switching sequence and dynamic output action potentials when the
circuit behaves in a spiking feature, in which $TS_2$'s switching-on time is behind $TS_1$'s
switching-off. Bottom panel: Schematic of TS devices' switching sequence and the
dynamic output action potentials when the circuit behaves in a bursting feature, in
which $TS_2$'s switching-on time is before $TS_1$'s switching-off and $TS_2$ switches twice.
(Simulation) **D** The neuron output presents two different firing features under

different input voltages. When the input is 0.7 V, the neuron fires in the spiking
feature, showing regular single-spike output, and when the input is 1.2 V, the neu-
ron fires in the bursting feature with two spikes per burst. (Experiment) **E** Under
triangle wave scanning with an amplitude of 1.3 V, the output of the neuron shows
a transition between the two firing features. When the input is -1.0 V, the output
of the neuron transitions from spiking to bursting. (Experiment) **F** Neuronal output
frequency as a function of input voltage. The firing cluster frequency of the neuron
increases linearly with increasing voltage. When the voltage input is 1.0 V, bursting
starts to appear in the neuron output, corresponding to a large bursting spike
frequency. (Experiment).

Biologists have found that, except for the spiking feature, AN$_2$ neurons
also prompt the high frequency bursting feature that dominates the
avoidance behavior of crickets under the ultrasonic stimulus, exhibit-
ing a mixed firing behavior[40]. The descending neurons decode the
"bursting" and "spiking" features in the mixed pattern via a selective
communication scheme, which includes an asynchronous decoding
function and ensures a timely response. In addition, the properties of
bursting—high frequency[41], high reliability[42], and strong stimulation to
postsynaptic neurons[43]—are advantageous when signals are required
to trigger responses very quickly. This behavior allows the sensory
system to make early decisions to take evasive actions, facilitating
reliable and rapid responses. These principles demonstrate the intel-
ligence of insects, which inspires our avoidance system design.

Inspired by the cricket's avoidance neural circuits, we develop an
SCNC comprising a LiDAR sensor and three H-H neurons based on
NbO$_2$ memristors to support the robot in performing obstacle avoid-
ance (Fig. 1B). The LiDAR sensor emulates the auditory receptor and
encodes the robot-obstacle distance into voltage signals. One H-H
neuron serves as the AN$_2$ interneuron to receive the input from the
sensor and encode the distance-related voltage into mixed firing pat-
terns, named mixed-coded neurons (MCN). A key point is that the H-H
interneuron features a probabilistic transition from spiking to burst-
ing, and the bursting ratio increases as the input intensity increases,
which is attributed to the intrinsic stochasticity of memristors. The
other two NbO$_2$ H-H neurons emulate the descending neurons that
extract the bursting and spiking features from the encoded mixed
pattern in the interneuron. This operation reliably emulates the
selective communication scheme observed in biology. According to
the functions, we denoted these two neurons bursting-detection
neuron (BDN) and spiking-detection neuron (SDN). Then, we utilize

the firing frequency of BDN and SDN to control the steering motors
and actuating motors, respectively. The firing frequency of the BDN
increases as the distance-related voltages increase, while the SDN is
vice versa. Thus, the closer the obstacle is to the robot, the larger the
steering angle and the slower the speed, as shown in Fig. 1C. Then, the
obstacle avoidance response of the robot is achieved by equipping it
with such an SCNC and computing with firing features.

## H-H neurons based on NbO$_2$ memristors
We build our SCNC inspired by the circuit structure in ref. 26. We first
construct an H-H neuron using NbO$_2$ memristors, as shown in Fig. 2A.
The H-H neuron circuit comprises two resistors ($R_1$, $R_2$), two capacitors
($C_1$, $C_2$), and two NbO$_2$-based threshold switching (TS) memristors with
d.c.-biased provided by two voltage sources ($E_1$, $E_2$). These two TS
devices represent the sodium and potassium ion channels in biological
neurons, respectively. The device is configured with a Pt/NbO$_2$/Ti
structure (see more fabrication details in Supplementary Fig. 1 and
Methods). To demonstrate H-H neuron circuits, we connect the TS
devices fabricated in the laboratory with off-the-shelf resistors and
capacitors via a printed circuit board (PCB) in this work. Figure 2B
shows the typical *I-V* curve of the two NbO$_2$ TS memristors and the
built model (see Methods for device models). The two devices show
nearly identical electrical performance, reducing the difficulty of set-
ting the H-H circuit parameters. Initially, the device is in a high resis-
tance state (HRS). When the voltage applied to the top titanium
electrode surpasses the threshold voltage ($V_{TH}$), the device switches
from HRS to a low resistance state (LRS) under a 400 μA compliance
current. Once the applied voltage is less than the hold voltage ($V_{Hold}$),
the device spontaneously switches back to the HRS. Due to the
electron-related switching mechanism, the device exhibits a fast

switching speed (-1 ns)[22] and high endurance (>10^10 cycles)[23], which is decent for neuronal applications. According to the spike numbers within one cluster, the spiking and bursting features are two basic features that enable the neurons' various firing behaviors. Thus, we illustrate the mechanisms of the generation of spiking and bursting action potentials through simulation, as shown in Fig. 2C. The top panel illustrates both TS devices' switching sequence and the dynamic output action potentials when the circuit behaves in a spiking feature. In this case, $TS_2$'s switching-on time is behind $TS_1$'s switching-off (see more detailed description of the switching procedures in Supplementary Fig. 2). When the circuit parameters change, $TS_2$ may switch on before $TS_1$ switches off. Under these circumstances, the voltage of the positive constant voltage source $E_2$ charges $C_1$ through the coupled $R_2$, resulting in a delay in the switching off of $TS_1$ normally, which results in twice switching on/off of $TS_2$ and then completing a bursting fire (see more details in Supplementary Fig. 3). Thus, the output pattern of the neuron circuit can be controlled by adjusting the circuit parameters to regulate the relative switching time between two TS devices. Changing the two capacitors' capacitances to control the firing features of the output in other literature[29] follows this principle. In addition, neuron circuits used for neuromorphic computing usually receive both positive and negative inputs. The H-H neuron circuit we construct is also capable of accepting negative inputs and the integration of both positive and negative inputs (see more details in Supplementary Fig. 4). Based on such an H-H neuron circuit with NbO₂ memristors, we successfully demonstrated 23 natural firing behaviors (see Supplementary Fig. 5 and Supplementary Table 2), as reported in other literatures[26,44].

For emulating AN₂ neurons, the spiking and bursting features should emanate from a fixed neuron. Thus, according to the studied mechanisms, we carefully set the circuit parameters such that the switching-off time of $TS_1$ and the switching-on time of $TS_2$ is close at a lower input. Under such parameters, when the input increases, the switching-off time of $TS_1$ and the switching-on time of $TS_2$ gradually approach each other, that is, the time difference $\Delta t$ between them gradually decreases until $TS_2$ switches on before the switching off of $TS_1$, which in turn prevents the switching off of $TS_1$, completing the transition from spiking to bursting. Thus, we achieve both "spiking" and "bursting" features in a fixed circuit through simulation (Supplementary Fig. 6, parameters are shown in Supplementary Table 3). Then, based on the simulation parameters, we conduct an experiment on the NbO₂-based H-H neuron, as shown in Fig. 2D. Under a low constant input voltage (0.7 V), the neuron fires in the spiking feature (fires in a single peak), while the neuron fires in the bursting feature (fires in two peaks per cluster) when the applied voltage increases to 1.2 V. The results show that through carefully modulating the parameters, our neuron circuit successfully emulates the behavior of AN₂ neurons. To further illustrate that the transition between spiking and bursting features is continuous and reversible (Fig. 2E), we applied a triangular pulse with an amplitude of 1.3 V as the input of the neuron circuit. At very low voltages (<-0.3 V), the neuron fires no spike. When the input gradually increases, the neuron begins to fire in spiking feature, and the interspike intervals gradually decrease with increasing the input voltage, signifying that the neuron's output frequency gradually increases. As the voltage continues to increase (up to -1.0 V), the neuron transitions to fire in bursting feature, and the intervals between bursts gradually decrease. When the amplitude of the triangular input is 0.9 V and 2.2 V (Supplementary Fig. 7), the output continues to satisfy the above requirements, which indicates a reversible firing feature transition behavior. To more clearly present the firing feature transition under different input voltages, we analyze the firing frequency evolution, as shown in Fig. 2F. We define every firing event between two adjacent refractory periods as a cluster, regardless of whether there are single or two peaks (spikes or bursts) in one cluster. The cluster frequency increases almost linearly with increasing the

input voltage, consisting of the effect of the input intensity on frequency in the spiking feature only (blue line). In addition, the firing frequency within a burst is particularly high when the bursting feature starts to appear in the firing pattern of the output, even up to 320 kHz, which also increases with increasing input voltage (yellow line). The characteristic of the high frequency of bursts makes it easy to distinguish them from the spikes in the spiking feature. The results show that our H-H neuron with intensity-driven feature transition behavior has the potential to be used for encoding emergencies, similar to the AN₂ neuron.

## Probabilistic feature transition in NbO₂-based H-H neurons

In biological AN₂ neurons, the transition between spiking and bursting features is not a binary event[14]. Rather, the firing patterns mix both features, in which the ratio of the bursting feature is related to the stimuli intensity and the stochasticity of ions moving through the channels. These characteristics provide the AN₂ neuron with powerful encoding capability. To emulate the mixed firing pattern with both features of the AN₂ neurons, we further studied to endow the intrinsic stochastic switching of TS devices into the feature transition processes. This occurs because the randomness of $V_{TH}$, $V_{Hold}$ and the resistance of the two devices affects the relative switching on or off timing of these two TS devices and hence may result in a probability transition behavior with mixed spiking and bursting rather than an abrupt transition. First, we explore the effect of the randomness of the $V_{TH}$ and $V_{Hold}$ on the output. Figure 3A shows 50 DC switching cycles of the two TS devices used under a 400 μA current. Clearly, the $V_{TH}$ and $V_{Hold}$ values of both devices fluctuate, which is helpful for enabling mixed firing features because the firing feature is affected by the relative switching on or off timing of these two TS devices. To further demonstrate that the switching cycle is stochastic, we extracted the $V_{TH}$ and $V_{Hold}$ values of both devices in 5000 cycles from an oscillator circuit in Supplementary Fig. 8. The results confirm that the distribution of $V_{TH}$ and $V_{Hold}$ of the two devices is highly uniform and satisfies a Gaussian distribution (Fig. 3B and Supplementary Fig. 9), proving the randomness of the devices and thus has great potential to enable the probabilistic transition of H-H neurons. According to the previous analysis, the voltage values of the two sources are redesigned to reside between $\frac{V_{TH} + V_{Hold}}{2}$ and $V_{TH}$, enabling the H-H neuron to work in a probabilistic transition state. We first study how the devices' stochasticity affects the output firing features through simulation. Figure 3C shows the output phase diagram of the neuron circuit when the variation is within a range satisfying the Gaussian distribution in Fig. 3B. When both $TS_1$'s $V_{Hold}$ and $TS_2$'s $V_{TH}$ are at the mean value, the neuron's output is in the critical state of the transition from spiking to bursting feature. When both $TS_1$'s $V_{Hold}$ and $TS_2$'s $V_{TH}$ decrease, the switching-off time of $TS_1$ is delayed and the switching-on time of $TS_2$ is advanced, which results in a pure bursting feature. Conversely, when both $TS_1$'s $V_{Hold}$ and $TS_2$'s $V_{TH}$ increase, the switching-off time of $TS_1$ is advanced, and the switching-on time of $TS_2$ is delayed. The output of the neuron circuit fires in a pure spiking feature. When the $V_{Hold}$ of $TS_1$ and the $V_{TH}$ of $TS_2$ change in opposite trends, the feature of the circuit's output is uncertain. Therefore, when the $V_{TH}$ and $V_{Hold}$ of the devices vary within a certain range, the circuit generates mixed firing features under a constant input. We note that instead of $TS_1$'s $V_{Hold}$ and $TS_2$'s $V_{TH}$, $TS_1$'s $V_{TH}$ and $TS_2$'s $V_{Hold}$ also affect the firing features, which we do not discuss here one by one individually. Similarly, we also explore the effect of the high and low resistance randomness of the device on the circuit output, as shown in Supplementary Fig. 10. It is shown that the combination of the randomness of $V_{TH}$, $V_{Hold}$, the high and low resistance value of the two devices results in the probabilistic transition behavior of the circuit output. Figure 3D shows the neuron's firing output under three different input intensities in the experiment. Under a low input voltage (0.9 V), the neuron fires with a pure spiking feature (top panel). When increasing the input voltage to 1.2 V, the neuron's

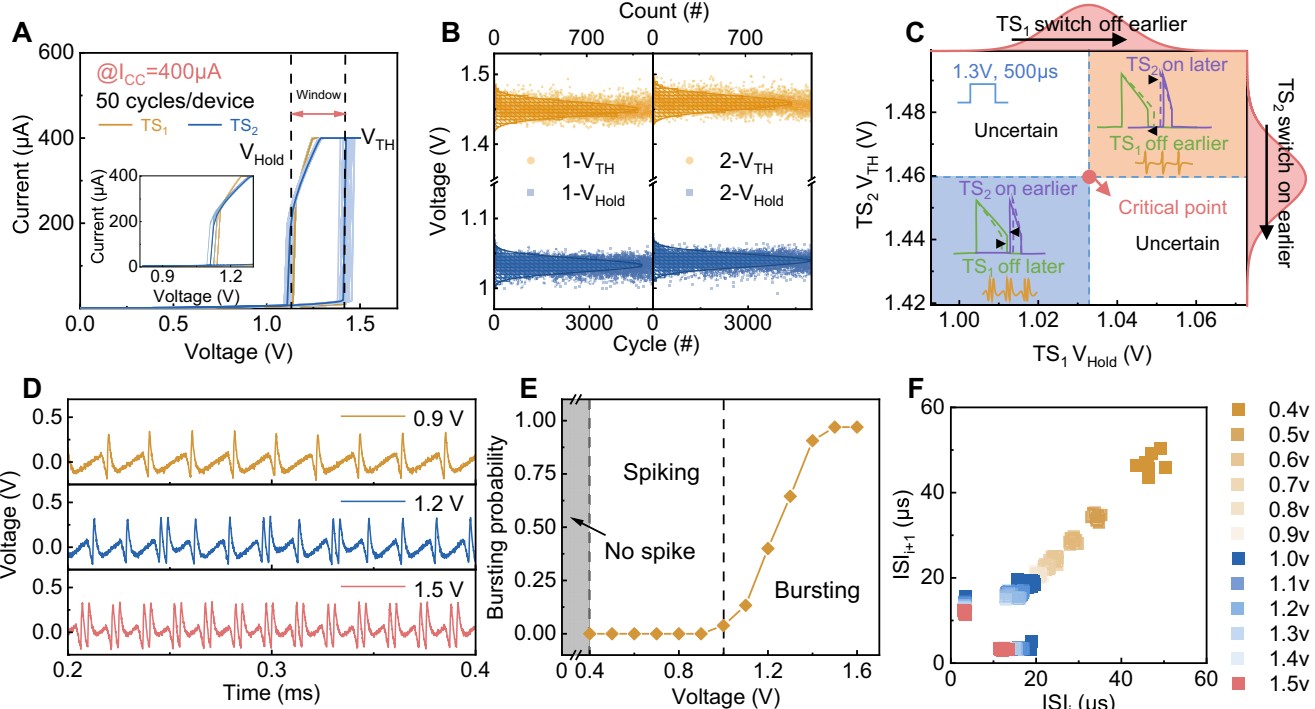

**Fig. 3 | Probabilistic transition behavior of the H·H neuron. A** The switching cycles of the two TS devices used for the H·H circuit, showing similar $V_{TH}$ and $V_{Hold}$ with random fluctuations. **B** The $V_{TH}$ and $V_{Hold}$ of the two devices show some randomness under 5000 cycles and satisfy the Gaussian distribution. **C** The output phase diagram of the neuron circuit when the $V_{Hold}$ of $TS_1$ and the $V_{TH}$ of $TS_2$ vary within a range satisfying the Gaussian distribution. When the $V_{TH}$ and $V_{Hold}$ of the devices are randomly changed in a certain range, the circuit can generate mixed firing features under constant input. The voltage values of the two sources are designed to be within $\frac{V_{TH}+V_{Hold}}{2}$ and $V_{TH}$. (Simulation) **D** The output of the neuron shows probabilistic transition behavior under different voltage inputs. At 0.9 V, the output of the neuron is in the spiking feature. At 1.2 V, the output of the neuron presents a mixed pattern of bursting and spiking, and at 1.5 V, the output is in the bursting feature. (Experiment) **E** The bursting probability of the neuron increases with the input voltage. (Experiment) **F** Joint interspike interval (JISI) scatter plots of the spike train with input from 0.3 V–1.5 V. (Experiment).

output contains both spiking and bursting features, showing probabilistic transition behavior (middle panel). When the input voltage is sufficiently high, the neuron fires with a pure bursting feature (bottom panel). The results show that after careful design, our neuron successfully emulates the mixed transition behavior in $AN_2$ neurons.

Figure 3E shows the calculation of the bursting probability of the neuron circuit under input voltage pulses (0.3 V–1.5 V, 500 μs) with a step of 0.1 V according to the data from Fig. 3D and Supplementary Fig. 11. To explore how the input intensity affects the feature transition processes, we define bursting probability as the ratio of bursting numbers to all cluster numbers during pulse application. When the voltage is low (<1.0 V), there is no burst in the output, which entirely comprises spiking, such the bursting probability is zero. When the input voltage increases to 1.0 V, some spikes transition to the bursts. With further increasing input, the bursting probability increases. When the input intensity is high enough, the influence of the input voltage increase exceeds the stochasticity induced by the TS devices, and the neuron fires continually in the bursting feature. To observe the properties of the firing pattern of the output under different inputs, we plot the interspike interval-time curve (see Supplementary Fig. 12). The results show that the interspike interval time exhibits irregular oscillations when the neuron fires in a mixed pattern, while it shows a constant or regular oscillation curve under a pure spiking or bursting case. To clearly illustrate the characteristics of the two firing features, the Joint Interspike Interval (JISI) scatter plots, which are also named interval return maps, of the spike trains are presented. The JISI plot is a classical stochastic spike train analysis method used to explore the relationship between adjacent $ISIs$[45–47]. We defined $ISI_i = t_n - t_{n-1}$, where $t_n$ is the time of the $n$th spike (see Supplementary Fig. 13). By plotting

the statistical distribution of adjacent $ISIs$, we could obtain the JISI plot and establish the serial correlation of adjacent points in the time series, which is helpful to judge and predict the firing pattern. Figure 3F shows the plotted JISI scatter plot of the neuron circuits under different input intensities. When the input is <1.0 V, the neuron fires in a pure spiking feature (yellow points), and the ISI points are gathered in the diagonal position, indicating that the interval between spikes is basically unchanged. When the input is no less than 1.5 V, the neuron fires with a pure bursting feature and the ISI points are distributed along the coordinate axis (pink points). The $ISIs$ of the first spikes in the bursts are closer to the $X$ axis, while the $ISIs$ of the second spikes are closer to the $Y$ axis. When 1.0 V < input < 1.5 V, the neuron fires in a mixed pattern, and $ISIs$ are located partly on the diagonal and partly in the region close to the coordinate axis (blue points). Thus, through the distribution of $ISIs$, we effectively distinguish whether bursting features are included in the output for different input intensities. These results demonstrate that the intrinsic stochasticity of the device enhances the encoding capability of H·H neurons and is a powerful source to enable feature-driven computing.

## Selective communication scheme in H·H neurons

To result in the execution of functional responses, the mixed firing features generated by the $AN_2$ neurons need to be further decoded by the descending neurons that control related motor neurons. Biologists have found that there is a selective communication mechanism between biological neurons, attributed to the subthreshold oscillation characteristics of biological neurons[48,49]. The descending neurons generate action potentials only when the input frequency matches their subthreshold oscillation frequency (see Supplementary Fig. 14 for

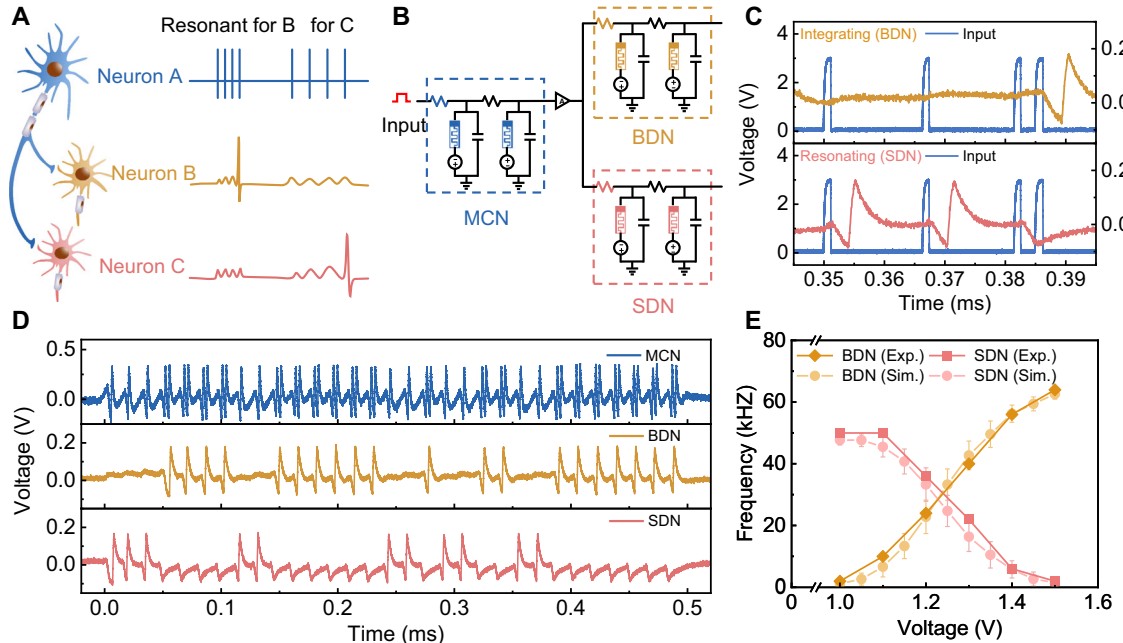

**Fig. 4 | Selective communication scheme between neurons. A** Schematic of selective communication of biological neurons. Postsynaptic neurons selectively transmit information from presynaptic neurons by responding to specific firing features with a resonance behavior. **B** Schematic of an artificial neural circuit emulating the biological selective communication circuit. The three memristive H-H neurons are configured with different circuit parameters. **C** The selective communication scheme in the memristive neural circuit. The resonance characteristic of the BDN neuron enables it to produce spike responses only to inputs with low frequency but not to inputs with high frequency. The integration property, on the other hand, makes the SDN neuron capable of producing spike output only for inputs with a high frequency. **D** The selective communication scheme of the neural circuit under continuous inputs. The SDN neuron with resonance characteristics can extract single spikes from the output of the mixed pattern of the MCN and generate spike responses, and the BDN neuron with integration characteristics extract bursts from the output of the MCN. **E** The firing frequencies of SDN and BDN under different input voltages on the MCN neuron. The simulation data is well-fitted with the experimental results. The error bars represent the standard deviation ($\sigma$) of each group.

more details). That is, descending neurons respond to those inputs with specific resonant frequencies. Specifically, neuron A sends spike trains with mixed firing features to neurons B and C, which have different subthreshold resonant frequencies, as shown in Fig. 4A. Neuron B, with a higher subthreshold oscillation frequency, resonates with the bursting feature at the higher frequency generated by neuron A and then generates action potentials but does not respond to the spiking feature. Neuron C, with a lower subthreshold oscillation frequency, responds with the opposite behavior. Consequently, descending neurons B and C can selectively respond to specific information in a mixed pattern from ascending neurons, which demonstrates intriguing inherent cognitive capabilities.

Inspired by the selective communication scheme between biological neurons, we configure another two NbO$_2$-based H-H neurons with different circuit parameters (see Supplementary Table 3) to process the mixed pattern sent by the AN$_2$-like H-H neuron. These two neurons are denoted BDN and SDN according to their functions. The neural circuits built upon these three neurons are intended to emulate the crickets escape neural circuit, as shown in Fig. 4B. The MCN encodes the input stimuli intensity, generating a mixed pattern that contains both spiking and bursting features, as discussed. The BDN fires action potentials after bursts from the MCN are detected, while the SDN neuron detects the spikes in the spiking feature. To improve the fan-out ability of the MCN, we adopted a comparator as the interface to transmit signals to the following descending neurons. Here, it should be noted that we design these three neurons under the same structure, which lowers the complexity of system design and enables us to build flexible systems when the neuron's components are configurable. Figure 4C shows the firing behaviors of the BDN and SDN under identical inputs with a mixed pattern. The BDN generates an action potential when the input is at a high frequency but does not respond to the low-frequency input (top panel of Fig. 4C), serving as an integrator[26,44]. While the SDN only responds to the low-frequency input (bottom panel of Fig. 4C), serving as a resonator[44,48]. These results show that the designed BDN and SDN could successfully detect bursting and spiking features, respectively.

To further demonstrate the selective communication function of the constructed neural circuit, we applied a 1.3 V voltage with a 500 μs duration on the AN$_2$-like H-H neuron and tested the output of the two descending neurons. The firing behaviors of these three neurons are shown in Fig. 4D. When the stimuli input is applied, the MCN fires with a mixed pattern due to the probabilistic transition (top panel). Under this condition, such a mixed spike train is transmitted to both the BDN and SDN. The BDN fires only when a bursting event occurs (bottom panel), while the SDN only responds to the spiking event (bottom panel). We also tested the output of the neural circuits under different input voltages, emulating the response of the neural circuit under different stimulus intensities, as shown in Supplementary Fig. 15. The higher the input voltage is, the higher the probability of bursting features, resulting in a higher fire frequency of the BDN and a lower frequency of the SDN. To more clearly present the computing process of the neural circuit, we plotted the average frequency output of BDN and SDN under different input voltages (the solid lines in Fig. 4E). To further illustrate the validity of the relationship between the input voltage and bursting/spiking firing frequency, we construct a mathematical device model with stochasticity of switching voltages and introduce it in the H-H neuron circuit. The simulated data is well matched with the experimental data, as shown in Fig. 4E (see more details in Supplementary Text 1 and Supplementary Fig. 16). Under an input voltage <1.0 V, the BDN neuron fires no spikes, indicating that there are no accidental triggers. When the input voltage increases to 1.0 V, the BDN starts to fire and the firing frequency increases as the input intensity increases,

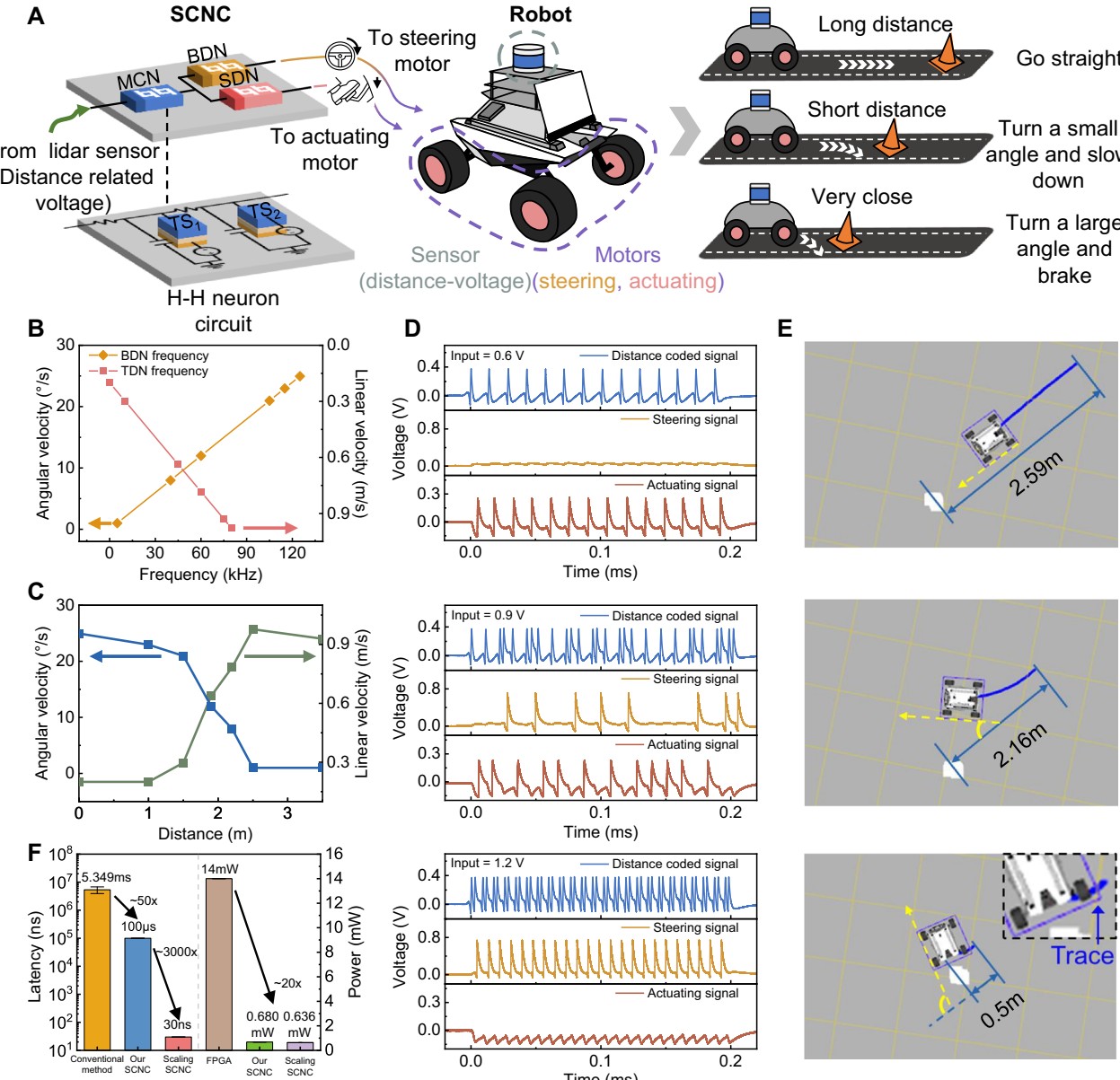

**Fig. 5 | Robot obstacle avoidance system driven by neuron firing features.**
**A** Schematic diagram of the robot avoiding an obstacle that appears at different distances. **B** The angular and linear velocities are linearly proportional to the BDN frequency and SDN frequency, respectively. A reasonable maximum linear velocity (0.9776 m/s) is set here. **C** An illustration of the relationship between variable distance range and the responses determined by SCNC, involving angular velocity and linear velocity. **D** The firing patterns of the MCN (blue), BDN (yellow) and SDN (red) are associated with the 'distance coded' signal, 'steering' signal and 'actuating'

signal, respectively. **E** The indoor trajectory recording of a robot vehicle under three typical features. The motion trace is in the cyan line, and the steering angle is in yellow. **F** Comparison of time latency between the obstacle avoidance scheme implemented by conventional methods (NVIDIA Jetson AGX Xavier), our physical SCNC and the SCNC after scaling. And a comparison of power consumption between the obstacle avoidance scheme implemented by FPGA, our physical SCNC and the SCNC after scaling.

corresponding to the increased bursting probability. The SDN's output behaves vice versa. When the input voltage is high enough, the BDN exhibits the highest frequency, while the SDN fires no spikes. These results show that the constructed neural circuit could successfully encode stimuli with mixed firing features and decode them through the intrinsic selective communication properties of neurons, faithfully emulating the functions of cricket escape neural circuits.

### Robot obstacle avoidance system with constructed SCNC
Mobile robots need to complete tasks intelligently in complex and changeable environments in a real-time and low-power manner[50]. When encountering an abrupt danger, such as an incoming high-speed

obstacle, the robot needs to react as quickly as possible to ensure its safety. Inspired by the reflex-like self-protection behavior of crickets for dodging predators, we present a strategy by equipping the SCNC on a robot to enable obstacle avoidance behavior in an emergency and achieve a rapid response. Our workflow is shown in Fig. 5A. LiDAR serves as the sensor, detecting obstacle distances. These distances are translated to voltages that serve as the input of the SCNC. The voltage is defined as inversely proportional to the distance. The MCN encodes the input voltage into mixed patterns. The SDN detects the spikes from the MCN, and the firing output is used to drive the actuating motors. Meanwhile, the BDN detects the bursts from the MCN, and the output is utilized to control the steering motors. The BDN and SDN output

frequencies are positively proportional to the angular velocity and linear velocity, respectively. The right part of Fig. 5A depicts three typical cases during driving. When the obstacle is quite far away, the MCN is driven by low voltage input to generate spike trains with pure spiking features, and thus, the SDN has the highest frequency output. In this case, the robot is actuated with the highest speed and moves straight (top panel). When an obstacle is relatively close, the input voltage enables the MCN to fire with mixed firing features, and the BDN neuron starts to fire to drive the steering motors. The robot decelerates and turns at a small angle (middle panel). When a sudden obstacle is very close, the robot executes an emergency response, namely, braking and rapid turning at a large angle, to remove the obstacle to a safe zone and prepare for the next straight move (bottom panel).

The linear mapping relationship between the firing frequency of BDN (SDN) and the angular velocity (linear velocity) during the experiment is shown in Fig. 5B. Figure 5C shows the angular velocity and linear velocity output of the robot at different distances. The closer the robot is to the obstacle, the smaller the linear velocity and the larger the angular velocity. Figure 5D shows the output results of the SCNC under three different distances. The distance information detected by the LiDAR is encoded as an input voltage and transmitted to the SCNC. The MCN, BDN and SDN in the circuit generate corresponding firing behaviors, which represent the "distance coded" signal, "steering" signal, and "actuating" signal, respectively, and are transmitted back to control the robot. The movement trajectory of the robot in the physical environment is recorded during the experiment to represent the linear velocity (length of the cyan trajectory) and angular velocity (angle marked by the yellow line) of the robot while driving. When the obstacle is quite far away (2.59 m), the decoded actuating signal has a high frequency response (top panel of Fig. 5D). In this case, the linear velocity reaches the maximum (0.9776 m/s), while the angular velocity is negligible because the steering signal contains few spikes. Therefore, the robot is quite safe and thus moves straight quickly (top panel of Fig. 5E). Once the obstacle appears within a preset dangerous distance range (2.16 m), the MCN fires in a mixed pattern (middle panel of Fig. 5D), representing certain danger is detected, and the steering signal and the actuating signal are used to drive obstacle dodging. The robot navigates around the obstacle with a moderate speed and angular velocity (middle panel of Fig. 5E). In contrast, if the robot notices that an abrupt incident occurs in close proximity (0.5 m) and does not have enough time and space for reaction, it rapidly decelerates and makes a sharp turn to ensure safety. Under such a circumstance, the distance coded signal exhibits almost all bursting (bottom panel of Fig. 5D), which gives rise to maximum angular velocity and minimum speed (bottom panel of Fig. 5E). Actually, these three responses in Fig. 5d, e are just discrete slices of continuous motion. In a continuous emergency response process, these movements are combined at different time durations to accomplish a complete obstacle avoidance course. More experimental details are presented in Methods and Supplementary Figs. 17–19. Supplementary Movies 1–4 show the whole emergent obstacle avoidance process.

Moreover, we calculate the average latency executed by the traditional CMOS computing unit in the NVIDIA Jetson AGX Xavier and compare it to that of our SCNC. After 500 trails of recording, the data shows that our SCNC alleviates the delay burden of robotic movement. It should be noted that the latency considered here is between receiving the distance signal and publishing the motion command. During the measurement, we processed the distance information as a voltage input with a pulse width of 200 μs, in which case the latency of the circuit is 200 μs. Here, we define the minimum inference time during which the statistical firing rate curve still obeys the relationship in Fig. 4E, as the minimum delay. When the pulse width is reduced to 100 μs, the statistical curve is still decent (see Supplementary Fig. 20). Therefore, we consider the minimum delay of the circuit to be 100 μs

under the current circuit parameters. Figure 5F illustrates the statistical results, and the latency of leveraging physical SCNC to directly drive the robot behavior is reduced by ~50 times. In fact, the minimum delay of the circuit is mainly determined by the two capacitors $C_1$ and $C_2$ in the circuit. When the capacitance of the device is reduced to an ~fF level ($C_1 = 500$ fF, $C_2 = 100$ fF of MCN)[22], the minimum delay of the circuit can be reduced to 30 ns (see Supplementary Fig. 20). At the same time, we conducted an evaluation of our obstacle avoidance approach on a Xilinx XCZU2CG-1SFVC784E (operating at a frequency of 25 MHz with a 1.8 V I/O standard), comparing its power consumption with our NbO$_2$ SCNC hardware realization scheme. The results show that the highest average power consumption of three H-H neurons in the SCNC is 0.680 mW, while the FPGA consumes an average of 14 mW, indicating that the power consumption of our NbO$_2$ SCNC is <5% of FPGA. The power consumption of the SCNC could be as low as ~0.636 mW when the capacitances are scaled, which achieves largely lower total energy consumption attributed to the smaller latency. In this case, each spike's energy consumption of the circuit can also be reduced to 1.06 pJ/Spike (see Supplementary Fig. 21 and Supplementary Table 4). These results indicate the great potential of our SCNC for the application of mobile intelligent robot emergent obstacle avoidance.

## Discussion

Neural circuits and their computational mechanics provide novel strategies for intelligent behavior control of robots. Inspired by the strategy of crickets to avoid natural enemies, we built an artificial SCNC for robot obstacle avoidance control. The SCNC is constructed with compact memristive H-H neurons and configured with a bionic selective communication scheme. Attributed to the memristor's intrinsic stochasticity, the H-H neurons exhibit mixed firing features in addition to the 23 reported firing behaviors. To demonstrate the feasibility of our SCNC for practical applications, we introduced it into a robot and successfully controlled the robot to perform obstacle avoidance. Compared with conventional robot obstacle avoidance algorithms conducted on a GPU platform, our SCNC features a >50× reduction in latency, which could be further improved by scaling the capacitances. When triggered in the face of danger, the neural circuit can help the robot to avoid at a faster speed.

The current H-H neuron circuits are constructed using a PCB board, resulting in a relatively large area that cannot be easily scaled down. A further study of on-chip integrated H-H neuron circuits is needed to better evaluate the neuron area and performance. A promising scalability comparison of memristor-based neurons and CMOS technology would be conducted, as illustrated in Supplementary Fig. 22. Only in this way can the advantages of neurons with memristors be decent claimed. In addition, by deploying two SCNCs on the robot, the left or right position of the obstacle relative to the robot can be judged instead of the distance, thus further controlling the turning direction. Similarly, increasing the number of SCNCs further improves the resolution of the robot in judging the orientation of the obstacle, which can help the robot achieve more accurate obstacle avoidance behavior. Moreover, the parameters of the current H-H neuron circuit are fixed, which makes the neurons lack plasticity. Further studies on endowing the neurons with plasticity are required, such as replacing the resistors in the circuit with configurable memristors or engineering configurable NbO$_x$ devices. Then such a configurable neuron with learning capability could adjust the memristor's resistance value after training. So that it could adapt to different dangerous distance ranges with more generality and thus work in various environments, such as indoor and outdoor, slow-motion obstacles, and high-speed obstacles. Through adjusting the memristor's resistance value after training, the neural circuit could adapt to different dangerous distance ranges and feature more generality. Furthermore, the LIDAR system we used requires an additional process to convert the distance signal into a

voltage signal through the CPU, which induces an additional delay and hinders our scheme from giving full play to its advantages to some extent. The direct connection between our neuronal circuit to resistive sensors (pressure or photoelectric sensors) can solve the problem and maximize the potential of the proposed solution, which deserves further exploration in the future. On the other hand, the lifetime of H-H neuron circuits is also worth addressing. To improve the lifetime of the neuron circuits, it is more important to improve the endurance of TS devices, which is undoubtedly one of the directions we should continue to pay attention to and study at the device levels. In practical applications, the impact of continuous input, pulse input and device degradation on neurons' firing behavior is also undoubtedly a topic worthy of further exploration. Nevertheless, our work, as a first step, brings spiking features into intelligent neural circuits, still has important implications, paving the way for real implementation of next-generation high-order brain-like intelligent systems in the future.

## Methods

### Fabrication of $NbO_2$ device

The $NbO_2$ threshold switching devices for the experimental demonstration were fabricated using e-beam evaporation, magnetron sputtering and lift-off. Ti (10 nm)/Pt (30 nm) bottom electrodes (BE) with a width of 5 μm were first grown by e-beam evaporation and patterned on a $Si/SiO_2$ substrate through a lift-off process. Then, a 50 nm-thickness of $NbO_x$ film is deposited on the BE at room temperature, by reactive sputtering with $NbO_2$ (Nb: $Nb_2O_5$ = 1:2) target in an atmosphere of Ar and $O_2$ ($O_2$: Ar = 0.12, at 3 mt). Finally, top electrodes of 5 μm width and 40 nm thickness (10 nm Ti/30 nm Pt) were patterned with a lift-off process perpendicular to the bottom electrode and deposited on the $NbO_2$ films by magnetron sputtering to form the 5 μm × 5 μm metal/$NbO_2$/metal crossbar junction. Initially, the virgin device is in a high resistance state (HRS). After forming, a $NbO_2$ channel forms in the $NbO_2$ layer and enables the device to exhibit TS characteristics. The SEM characterization of the device is shown in Supplementary Fig. 1.

### Device measurement

For electrical measurement, the DC test of the $NbO_2$ device was performed on an Agilent B1500A semiconductor parameter analyzer. In pulse tests, a Keysight 81160 A pulse generator served as the power source, and a Keysight Infinii Vision MSO-X 3104 T oscilloscope was chosen to monitor electrical pulse signals.

### Neuron circuit implementation

The H-H neuron circuits were constructed by connecting two $NbO_2$ devices with resistors and capacitors via a printed circuit board (PCB). The WGFMU module of the Agilent B1500A was used to apply pulses to neuron circuits. The inputs of the neuron circuit, $V_{Na}$ and $V_K$ were monitored using the MSO-X 3104 T oscilloscope, and the voltage bias was applied to the two $NbO_2$ devices using the Keithley 2230 G constant voltage source. The SCNC contained three H-H neuron circuits based on three pairs of $NbO_2$ devices and a comparator as shown in Supplementary Fig. 23.

### Device and circuit simulation

The LT SPICE model was used to simulate the $NbO_2$ switching dynamics and the mechanism of the spiking and bursting features (Fig. 2C, Fig. 3C and Supplementary Fig. 5). Supplementary Table 5 illustrates the parameter values of the $NbO_2$ device for simulation. All the H-H neurons used the same $NbO_2$ device model and only varied the input values. The $NbO_2$ device model used to simulate the randomness of the threshold voltage and the H-H model used to simulate the probabilistic transition feature are shown in Supplementary Information 1.

### Robotic obstacle avoidance control

The nearest obstacle monitored within the detection degree range from −90° – 90° (toward the front half of the robot) is viewed as the emergency that the robot should escape. Point cloud data perceived by Velodyne 16 LiDAR are qualified for both capturing distance information of emergency and mapping of the environment in Fig. 5E. The distance information detected by LiDAR is processed into voltage pulses with a pulse width of 500 μs and transmitted to the SCNC. Toward the goal of biological plausibility, as the emergency occurs closer, the neural circuit ought to be stimulated more intensely, and thus, we set a larger voltage input under the condition of a smaller obstacle distance. The output of the SCNC (Supplementary Fig. 13) is then linearly decoded as the angular and linear velocity. In the motion trace record experiment (Fig. 5E and Supplementary Movies 1–3), the locations where the robot sets off and the adverse obstacle appears are fixed, but the appearance times of obstacles are different under three situations for the sake of monitoring the distance-motion response relationship. NVIDIA Jetson AGX Xavier is used for computing tasks in our mobile robot. We then measured the simulation latency of the conventional hardware that completes sensorimotor control, which is replaced by SCNC in our experiment, i.e., the module between the LiDAR scan node and mobile robot base node, using the *Python time* module to obtain the execution duration on NVIDIA Jetson AGX Xavier. The average delay is >5000 μs in 100 measurements. In comparison, our neural circuit's response time is within 100 μs in the physical test, which is ~50 times faster.

## Data availability

Source data are provided with this paper.

## Code availability

Supplementary code is provided with this paper.

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

## Acknowledgements

This work was supported by the National Natural Science Foundation of China (Nos. T2293732 (Q.L.), 62374040 (X.Z.), and 62104044 (X.Z.)), the Strategic Priority Research Program of the Chinese Academy of Sciences (No. XDB44000000 (M.L.)), the Zhejiang Lab Open Research Project (No. K2022PF0AB01 (X.Z.)), and the project of MOE innovation platform.

## Author contributions

Y.Y., X.Z. and Q.L conceived the concept and designed the experiments. Y.Y., P.C. and J.Z. fabricated the $NbO_2$ devices and established the LT spice model of the devices. Y.Y., X.Z. and Y.W. performed electrical tests on the device and the H-H neuron circuits. Y.Y established the LT spice model of the H-H neuron circuits. F.Z. constructed the model of the $NbO_2$ device with random $V_{TH}$ and $V_{Hold}$ and the H-H circuit model with probabilistic transition behavior and implemented the control of the robot obstacle avoidance system by the SCNC. Y.Y., X.Z., F.Z., Y.D., L.C., C.L., H.J., Z.W., P.L., T.S., M.W. and Q.L. contributed to the analysis and interpretation of results. Y.Y., F.Z. and X.Z. wrote the manuscript with input from all authors. X.Z., Q.L., N.X. and M.L. supervised the whole project.

## Competing interests

The authors declare no competing interests.
