## [Peer Review File · Nature Communications]

REVIEWER COMMENTS

Reviewer #1 (Remarks to the Author):

The authors in this manuscript developed a neural circuit that has a neuron to convert analog voltage input signal into spiking with either tonic or bursting features, and then two other types of neurons to detect and differentiate such features, which can serve as the indicators of distance to an object and thus information for decision making, namely accelerate or slow down and turn a robot. The interesting part is that all the three types of neurons are based on the same circuit consisting of two memristors, two resistors and two capacitors, but with different parameters. The selective communication between the MCN and BDN/SDN is novel. The data are solid, device performance is great, and the demos are impressive. Therefore, I recommend it to be accepted after addressing the following questions.

- 1) More details are needed for the NbO₂ deposition, for instance, was it reactive sputtering or sputtering with NbO₂ or Nb₂O₅ target? In pure Ar or with Ar+O₂?
- 2) What was the initial composition of the sputter-deposited NbO_x film and what happens to the film composition during the electroforming step?
- 3) What is the role of Ti electrode? Why not other type of metals?
- 4) For the experimental demos, how many pairs of NbO₂ devices were used, only one or 3 pairs as shown in Fig. 4 and Supplementary Fig. 20?
- 5) In each burst, there are only two spikes. How to obtain more spikes per burst?
- 6) The title claims scalable memristor neurons while only micron size devices are shown.
- 7) It would be nice to have more comparisons with purely CMOS solutions to show the advantages of this approach, in addition to response latency.

Reviewer #2 (Remarks to the Author):

In this manuscript, the authors report on a neuromorphic circuit composed of three heterogeneous neurons inspired by the avoidance neural circuits of crickets. A single neuron is composed of two Threshold Switching devices connected in parallel with two different capacitors. As a demonstration, the

authors propose to utilize the circuit connected to a LiDAR sensor for obstacle avoidance in robots. This approach has the advantage of low latency compared to conventional platforms.

First, I have the impression that most of the results are primarily simulation-based, relying on experimental data from a single device characterized under DC conditions (Fig. 2b and Figs. 3a, b). In contrast, all the figures depicting the neuron behavior (Figs. 2c, d, e, f and Fig. 3d, e, f) appear to be based on hardware-calibrated simulations. In the method section, it is clear that the neuron has not been fabricated, but rather, two Threshold Switching (TS) devices have been connected with resistors and capacitors via a PCB. However, it remains unclear where the measurement data obtained with this setup are located. It should be written for each figure is generated by the measurement of the PCB circuit, hardware calibrated measurements, or measurements on a single device. It would be more informative to present the mechanisms of generating spiking and bursting action potentials through experimental data.

The idea of using a single Threshold Switching device with a capacitor, the TS/C structure, to implement a spiking neuron is not novel, and a comparison with previously published works is absent. The novelty here lies in the coupling of two 1TS/C structures. However, the TS/C structure has some limitations that warrant discussion:

1. The proposed neuron can only integrate positive currents. The proposed application does not require the integration of negative currents. However, to achieve high-precision spiking neural networks, it is essential to accommodate both positive and negative weights. This implies that neurons should be capable of integrating both types of currents. Therefore, the proposed neuron may face challenges in its implementation for spiking neural networks.
2. The parameters of the neuron are fixed, defined by the values of the resistances and capacitors. In contrast, biological neural networks typically feature neurons with heterogeneous parameters that can adapt through homeostatic plasticity rules.
3. Volatile devices, such as TS, used for neurons, are not suitable for implementing synaptic weights, which require non-volatile memories. In the hardware implementation of neural networks, the primary challenge lies in synapses, rather than neurons, as there are orders of magnitude more synapses than neurons. Fabricating configurable memristors and TS devices requires additional masks, which can significantly increase costs.

A comparison with CMOS-based analog DPI neurons is essential [1, 2]. It is important to highlight the advantages of the proposed solution. Additionally, a comparison of power consumption would be beneficial.

Since the input voltage is not a spike but rather a constant voltage, it raises questions about the device's reliability and power consumption. For example, with an input voltage of 0.8V, which corresponds to

approximately 80kHz (as shown in Fig. 2f), there is a spike every 12.5 microseconds. Assuming an endurance of $1e10$ cycles, the maximum lifetime would be approximately 35 hours.

The impact of intrinsic stochastic switching on TS devices is thoroughly studied and leveraged. However, the effect of conductance device variability from device to device and cycle-to-cycle is not explored.

Which circuit is used to convert the LiDAR output into an input for the neuromorphic system? Additionally, what is the latency of a LiDAR system in measuring a distance and converting it into input voltage for the neuromorphic circuit?

[1] Arianna Rubino, Melika Payvand, Giacomo Indiveri Ultra-Low Power Silicon Neuron Circuit for Extreme-Edge Neuromorphic Intelligence, IEEE International Conference on Electronics Circuits and Systems (ICECS), 2019

[2] M. V. Nair and G. Indiveri, "An ultra-low power sigma-delta neuron circuit," in 2019 IEEE International Symposium on Circuits and Systems (ISCAS), May 2019, pp. 1–5.

We thank all the reviewers for their precious time and constructive comments on our manuscript. We revised both the main text and the supplementary information (SI) accordingly.

The changes we made are summarized as follows:

1. We provided more detailed process parameters for the fabrication of NbO₂ devices in method.
2. We provided the detailed description of the device numbers used by SCNC in Method.
3. Added detailed description to distinguish the data obtained through experiment or simulation in the legends of Figures 2 and 3.
4. We provided the power consumption comparison of FPGA, our SCNC, and SCNC after scaling in Figure 5f, and add corresponding descriptions in the main text.
5. Added a comparison of our work with literature reported artificial neuron based on memristors, the results are shown in the added Supplementary Table 1.
6. Added a comparison of our work with literature reported artificial neuron based on CMOS, the results are shown in the added Supplementary Table 4. Added a reference paper on page 3 when describing CMOS-associated H-H neurons.
7. We provided the effect of the devices' high and low resistance variations on the output of the constructed H-H circuit, and added corresponding descriptions in the main text, the results are shown in the added Supplementary Fig. 8.

In the following point-to-point response, the original comments are in black fonts, and our responses are in blue fonts. Changes in the revised main text and SI are highlighted

in yellow color.

One by one Response to Reviewers' comments:

Reviewer #1 (Remarks to the Author)

The authors in this manuscript developed a neural circuit that has a neuron to convert analog voltage input signal into spiking with either tonic or bursting features, and then two other types of neurons to detect and differentiate such features, which can serve as the indicators of distance to an object and thus information for decision making, namely accelerate or slow down and turn a robot. The interesting part is that all the three types of neurons are based on the same circuit consisting of two mott memristors, two resistors and two capacitors, but with different parameters. The selective communication between the MCN and BDN/SDN is novel. The data are solid, device performance is great, and the demos are impressive. Therefore, I recommend it to be accepted after addressing the following questions.

Response: We thank the referee for the positive comments on the significance of our work. Our responses to the comments one by one are shown as follows.

1) More details are needed for the NbO₂ deposition, for instance, was it reactive sputtering or sputtering with NbO₂ or Nb₂O₅ target? In pure Ar or with Ar+O₂?

Response: We thank the referee for this comment. The 50nm-thickness of NbO_x film is deposited by reactive sputtering with NbO₂ (fabricated with Nb:Nb₂O₅ = 1:2) target in an atmosphere of Ar and O₂ (O : Ar = 0.12, at 3 mt). To clear this, we added descriptions in the method part. "Ti (10 nm)/Pt (30 nm) bottom electrodes (BE) with a width of 5

μm were first grown by e-beam evaporation and patterned on a Si/SiO₂ substrate through a lift-off process. Then, a 50nm-thickness of NbO_x film is deposited on the BE at room temperature, by reactive sputtering with NbO₂ (Nb:Nb₂O₅ = 1:2) target in an atmosphere of Ar and O₂ (O₂ : Ar = 0.12, at 3 mt).”

2) What was the initial composition of the sputter-deposited NbO_x film and what happens to the film composition during the electroforming step?

Response: We thank the referee for this comment. We deposited NbO_x films on SiO₂ films and then carried out X-ray photoelectron spectroscopy (XPS) analysis on films. Considering that the surface of the films would be oxidized and contaminated in the atmosphere, all samples were surface etched for 7 nm using Ar⁺ sputtering in the XPS chamber before collecting information. From the XPS results we can conclude, as shown in Fig. R1a, the initial composition of the sputter-deposited NbO_x film includes NbO (3.72%), NbO₂ (27.30%) and Nb₂O₅ (68.98%).

TEM analysis shows that the film is in an amorphous state before forming (Fig. R1b). After one step forming process, the crystalline phase of NbO₂ in the NbO_x film formed (Fig. R1c) to constitute the conductive channel. The results are consistent with those reported phenomenon^{1,2}.

Fig. R1 (a) XPS analysis of NbO_x films grown on SiO_2 substrate. (b) The TEM characterization of the devices before and (c) after electroforming process. The result shows that the crystalline phase of NbO_2 in the NbO_x film formed during forming process.

3) What is the role of Ti electrode? Why not other type of metals?

Response: We thank the referee for this comment. The function of the Ti film in the bottom electrode Ti/Pt is to be used as an adhesive layer, so that the Pt electrode can adhere well to the wafer substrate. The function of the Ti film in the top electrode Ti/Pt is mainly to be used as an active electrode. As previously reported in the literatures,

active electrodes contribute to the formation of crystalline phase NbO_2^{2-4} . Moreover, it is reported that oxide barrier layer formed by the active electrode blocks the movement of oxygen and further oxidation of the electrode, which can improve the uniformity of the device⁵. In addition to Ti electrode, other active electrodes such as TiN^{5-7} , Ru^5 , W^3 , $\text{Nb}^{3,8}$ can also be used.

4) For the experimental demos, how many pairs of NbO_2 devices were used, only one or 3 pairs as shown in Fig. 4 and Supplementary Fig. 20?

Response: We thank the referee for this comment. For the data in Figure 4, we utilized two pairs of NbO_2 devices. One pair of NbO_2 devices was used to implement the functionality of mixed-coded neuron, while another pair of NbO_2 devices was used to achieve two neurons with the functionality of bursting and spiking-detection by varying R_1 conditions. In other words, bursting and spiking-detection neuron in Fig.4 use the same pair of NbO_2 devices, but the resistance value of R_1 in the circuit is changed. This process was employed to validate the feasibility of selective communication within the neural circuit. While in Fig. 5 of the main text and Supplementary Fig. 20, we utilized three pairs of NbO_2 devices to construct neural circuits and demonstrate their applications.

To clear this, we add descriptions in the methods: “The SCNC contained three H-H neuron circuits based on three pairs of NbO_2 devices and a comparator, as shown in the Supplementary Fig. 20.”

5) In each burst, there are only two spikes. How to obtain more spikes per burst?

Response: We thank the referee for this comment. The most effective way to obtain more spikes per burst experimentally is to resize C_1 and C_2 to make a larger ratio of C_1/C_2 in the circuit. We explored the effect of changes in R_1 , R_2 , E_1 , and E_2 on the output behavior of neuronal circuits through LT spice simulations, as shown in Fig. R3a-d. The initial circuit parameters are $C_1 = 2.2$ nF, $C_2 = 470$ pF, $R_1 = 25$ k Ω , $R_2 = 4$ k Ω , $E_1 = 2.82$ V, and $E_2 = 2.73$ V. When exploring the influence of parameters on the circuit output, only one circuit parameter and the voltage input are changed, and the other parameters are fixed. The results show that the uncoupled parameter R_1 serves as a voltage-divider, inducing regular and linear states transition boundaries between different firing behaviors. While the states transition boundaries under coupled parameters (R_2 , E_1 & E_2) are nonlinear due to their influence on charge/discharge of C_1 & C_2 . Moreover, according to the area represented by each firing behavior in the figure, it can be concluded that the circuit conditions for transition from tonic spiking to tonic bursting (two spikes) are not stringent. For example, from Figure 3a, it can be observed that when other component parameters remain constant, the transition in firing behavior can be observed with increasing input when $R_1 > 10$ k Ω . In contrast, achieving tonic bursting (three spikes) is very demanding. During experimental testing, due to the randomness of device thresholds and parasitic capacitance, it is challenging to observe the continuous transition from tonic spiking to burst-2 and then to burst-3 as the input increases. As mentioned in the main text, the firing behavior of the output mainly depends on the ratio of two devices switch on and off times. The parameter that has the greatest impact on this is the two capacitors' value. Therefore, while keeping other

circuit component parameters fixed, we varied the two capacitance values separately and observed the circuit's output behavior. As shown in Fig. R3e-f, when C_1 increases, or when C_2 decreases, or in other words, when the ratio between them increases, the circuit is more likely to produce multi-spikes bursting, and the larger the ratio, the more spikes within each bursting. This is because the opening time of TS_1 is long enough (due to a large C_1), allowing TS_2 to switch on and off multiple times (due to a small C_2). Therefore, the most effective way to get more spikes per burst is to increase the ratio of C_1/C_2 .

Fig. R3 Effect of circuit element parameter adjustment (a) R_1 , (b) R_2 , (c) E_1 , (d) E_2 , (e) C_1 and (f) C_2 on circuit output firing behavior. No spike-1: Both TS devices don't switch on. No spike-2: Both TS devices don't switch off. Fast spiking: TS₂ switch on and off continuously with TS₁ always in on-state. Burst-2/3/4/5: Two/three/four/five spikes within one burst. Subthreshold oscillation (Sub): Since TS₂ does not switch on, the neuron oscillates below the threshold. Tonic spiking: Neurons fire regularly with

single spike.

6) The title claims scalable memristor neurons while only micron size devices are shown.

Response: We thank the referee for this comment. Although the devices we used in this work are NbO₂ TS devices with an area of 5μm×5μm, the previous work⁶ in our group has reported NbO₂ devices with an area of 60nm×60nm. The scaled devices also have stable threshold switching characteristics and can be used to construct H-H neuron circuits.

7) It would be nice to have more comparisons with purely CMOS solutions to show the advantages of this approach, in addition to response latency.

Response: Many thanks for your suggestion. We conducted an evaluation of our obstacle avoidance approach on a Xilinx XCZU2CG-1SFVC784E (operating at a frequency of 25MHz with a 1.8V I/O standard), comparing its power consumption with our NbO₂ SCNC hardware realization scheme. The whole processing flowchart is shown in Fig. R4. To ensure a fair comparison, we calculated the energy consumption in ‘Phase 3’, involving the computing of velocity and turning angles based on obstacle distance. The results show that the highest average power consumption of three H-H neurons in the SCNC is 0.680mW, while the FPGA consumes an average of 14 mW, indicating that the power consumption of our NbO₂ SCNC is less than 5% of FPGA. It should be noted that the power consumption of the SCNC almost does not decrease as

the capacitance is scaled, which is due to the fact that the device is switched on more times and for less time when the capacitance is reduced. However, when the capacitor scaled, it takes less time for SCNC to control, so the energy consumption will still be reduced. Therefore, our SCNC approach demonstrates a power-efficient advantage except for lower latency. Correspondingly, we added the power consumption comparison in Fig. 5f and the description in main text. “When the capacitance of the device is reduced to an ~fF level ($C_1 = 500$ fF, $C_2 = 100$ fF), the minimum delay of the circuit can be reduced to 30 ns (see Supplementary Fig. 18). At the same time, we conducted an evaluation of our obstacle avoidance approach on a Xilinx XCZU2CG-1SFVC784E (operating at a frequency of 25 MHz with a 1.8V I/O standard), comparing its power consumption with our NbO₂ SCNC hardware realization scheme. The results show that the highest average power consumption of three H-H neurons in the SCNC is 0.680 mW, while the FPGA consumes an average of 14 mW, indicating that the power consumption of our NbO₂ SCNC is less than 5% of FPGA. It should be noted that the power consumption of the SCNC almost does not decrease as the capacitance is scaled (~0.7 mW), which is due to the fact that the devices switch on more times when the capacitor is scaled. However, when the capacitor scaled, the latency of SCNC is smaller, which means the work time of SCNC is shorter, so the total energy consumption is reduced.” We also updated Fig. 5f to illustrate the comparison of power consumption, as shown in Fig. R5 (Fig. 5f).

Fig. R4. Detailed flowchart of the entire obstacle avoidance. In Phase 1, Ridar senses and sends data to an industrial computer and then the point cloud data in a packet is compressed and decoded to obtain distance information. In Phase 2, the distance is mapped to a voltage for the convenience of input to a SCNC behavior model. In Phase 3, the SCNC behavioral model supports simulation and output velocity instructions for following motor control.

Fig. R5. Comparison of time latency between the obstacle avoidance scheme

implemented by conventional methods (NVIDIA Jetson AGX Xavier), our physical SCNC and the SCNC after scaling. And comparison of power consumption between the obstacle avoidance scheme implemented by FPGA, our physical SCNC and the SCNC after scaling.

Reviewer #2 (Remarks to the Author)

In this manuscript, the authors report on a neuromorphic circuit composed of three heterogeneous neurons inspired by the avoidance neural circuits of crickets. A single neuron is composed of two Threshold Switching devices connected in parallel with two different capacitors. As a demonstration, the authors propose to utilize the circuit connected to a LiDAR sensor for obstacle avoidance in robots. This approach has the advantage of low latency compared to conventional platforms.

1. First, I have the impression that most of the results are primarily simulation-based, relying on experimental data from a single device characterized under DC conditions (Fig. 2b and Figs. 3a, b). In contrast, all the figures depicting the neuron behavior (Figs. 2c, d, e, f and Fig. 3d, e, f) appear to be based on hardware-calibrated simulations. In the method section, it is clear that the neuron has not been fabricated, but rather, two Threshold Switching (TS) devices have been connected with resistors and capacitors via a PCB. However, it remains unclear where the measurement data obtained with this setup are located. It should be written for each figure is generated by the measurement of the PCB circuit, hardware calibrated measurements, or measurements on a single divide. It would be more informative to present the mechanisms of generating spiking

and bursting action potentials through experimental data.

Response: We thank the referee for this comment. We are sorry for any confusion caused. In fact, most of the results in our work is based on the experiment, except for Figures 2c and 3c that are based on the simulation data of hardware-calibration for clearly clarifying the causes of firing behaviors. This is because the current flowing through the two devices is better to understand the time sequence of switching on and off of the two TS devices, but it is hard to be observed through experiment. Figure 2d, e, f and Figure 3d, e, f in the main text are experimental data obtained by constructing H-H neurons on PCB, instead of simulation data. To avoid confusion, we marked in the legend whether each data graph was obtained by simulation or experiment.

“Fig. 2 | H-H neuron circuit based on NbO₂ memristors and the firing behaviors.

a, The H-H neuron consists of two resistors (R_1 , R_2), two capacitors (C_1 , C_2), and two TS devices (TS_1 , TS_2) with opposite bias voltages provided by two voltage sources (E_1 , E_2). **b,** Typical volatile threshold switching behavior of two TS devices used in the H-H neurons. The two TS devices exhibit similar V_{TH} and V_{Hold} . **c,** Top panel: Schematic of TS devices' switching sequence and the dynamic output action potentials when the circuit behaves in a spiking feature, in which TS_2 's switching-on time is behind the TS_1 's switching-off. Bottom panel: Schematic of TS devices' switching sequence and the dynamic output action potentials when the circuit behaves in a bursting feature, in which TS_2 's switching-on time is before TS_1 's switching-off and TS_2 switches twice.

(Simulation) d, The neuron output presents two different firing features under different

input voltages. When the input is 0.7 V, the neuron fires in the spiking feature, showing regular single-spike output, and when the input is 1.2 V, the neuron fires in the bursting feature, containing two spikes within each burst. (Experiment) e, Under triangle wave scanning with an amplitude of 1.3 V, the output of the neuron shows a transition between the two firing features. When the input is approximately 1.0 V, the output of the neuron transitions from spiking to bursting. (Experiment) f, Neuronal output frequency as a function of input voltage. The firing cluster frequency of the neuron increases linearly with increasing voltage. When the voltage input is 1.0 V, bursting starts to appear in the neuron output, corresponding to a large bursting spike frequency at this time. (Experiment)”

“Fig. 3 | Probabilistic transition behavior of the H-H neuron. a, The switching cycles of the two TS devices used for the H-H circuit, showing similar V_{TH} and V_{Hold} with random fluctuations. **b,** The V_{TH} and V_{Hold} of the two devices show some randomness under 5000 cycles and satisfy the Gaussian distribution. **c,** The output phase diagram of the neuron circuit when the V_{Hold} of TS₁ and the V_{TH} of TS₂ vary within a range satisfying the Gaussian distribution. When the V_{TH} and V_{Hold} of the devices are randomly changed in a certain range, the circuit can generate mixed firing features under constant input. The voltage values of the two sources are designed to be within $\frac{V_{TH}+V_{Hold}}{2}$ and V_{TH} . (Simulation)**d,** The output of the neuron shows probabilistic transition behavior under different voltage inputs. At 0.9 V, the output of the neuron is in spiking feature. At 1.2 V, the output of the neuron presents a mixed pattern of bursting and spiking, and at 1.5 V, the output is in the bursting feature. (Experiment) **e,** The

bursting probability of the neuron increases with the input voltage. (Experiment) f, Joint interspike interval (JISI) scatter plots of the spike train with input from 0.3 V to 1.5 V. (Experiment)”

2. The idea of using a single Threshold Switching device with a capacitor, the TS/C structure, to implement a spiking neuron is not novel, and a comparison with previously published works is absent. The novelty here lies in the coupling of two 1TS/C structures. However, the TS/C structure has some limitations that warrant discussion.

Response: We thank the referee for this comment. We agree with the referee that TS/C structures have been widely used in the realization of artificial neurons. In particular, there is a lot of work to construct LIF neurons based on one TS/C structure to build neural networks^{1,7,9}. However, the LIF neurons have certain limitations in terms of biological plausibility and cannot simulate the rich firing behavior of biological neurons, which limits the function of neurons in the network to a certain extent. Two TS/C coupled structures are adopted in Pickett’s work¹⁰ to realize a variety of firing behaviors of biological neurons, which provides us with a new idea. Inspired by this, there are also some work using two TS/Cs coupling structure to construct H-H neurons¹¹⁻¹². However, these works can only achieve few of firing behaviors, or do not utilize the computing capability of these firing behaviors, which has certain limitations. As stated by the referee, we adopted two TS/C coupled structures to construct neuron circuits and realized 23 firing behaviors of biological neurons. Moreover, for the first time, we constructed a three-neuron selective communication circuit that takes advantage of the computing capability to make decisions between different firing behaviors. Thus, we

believe this work provides an innovative way to compute with spiking features and has a great potential for building high-level intelligent machines.

Certainly, as mentioned by the referee, the TS/C structure does have some limitations. Firstly, the large area caused by capacitance may result in poor integration density of the circuit. This can be addressed by using parasitic capacitance to replace larger capacitances in the circuit. Additionally, the fixed capacitance values may lead to poor reconfigurability of the neuron circuit, which is also a common challenge in analog circuits designed for customized functionalities. In the future, we can improve the reconfigurability of the neuron circuit by replacing fixed capacitors with memcapacitors or optimizing the fabrication process to create threshold-adjustable TS devices. Moreover, the current TS devices in the TS/C structure exhibit relatively small high resistance value and thus induce a low RC time constant, which make it can only be increase the capacitance values in the TS/C structure to obtain a low firing frequency. Future efforts can focus on device optimization to reduce the leakage current while maintaining device switching speed, thereby expanding the range of the neuron's output frequency. Although the TS/C structure has certain limitations, it is still a highly efficient method to achieve high biological plausibility neuron circuits.

To clearly illustrate the difference between our work and other works, we have added the table in the supplementary information and the description in the main text on page 3 as follow: “However, current works typically focus on the emulation of neurons’ firing behaviors, and the exhibition of firing features’ computational capability remains an open question (see Supplementary Table 1).”

Supplementary Table 1. | Comparison between our work and literature reported artificial

neurons based on memristors

Neuron model	LIF	H-H	H-H	H-H	H-H
References	[1,7,9]	[10]	[11]	[12]	Our work
Functional material	NbO _x	NbO _x	VO ₂	VO ₂	NbO ₂
Number of achieved firing features	4 (All-or-nothing, refractory period, class 1 excitable, tonic spiking)	4 (All-or-nothing, refractory period, tonic spiking, tonic bursting)	23	5 (All-or-nothing, refractory period, tonic spiking, tonic bursting, spike frequency adaptation)	24 (23 kinds firing behaviors in [10] and probability switch)
Application	Yes	No	No	Yes	Yes
Firing features utilized in the application	1 (class 1 excitable)	-	-	1 (spike frequency adaptation)	5 (tonic spiking, tonic bursting, probability switch, resonator, integrator)

3. The proposed neuron can only integrate positive currents. The proposed application does not require the integration of negative currents. However, to achieve high-precision spiking neural networks, it is essential to accommodate both positive and negative weights. This implies that neurons should be capable of integrating both types of currents. Therefore, the proposed neuron may face challenges in its implementation for spiking neural networks.

Response: We thank the referee for this comment. We agree with the referee that a

spiking neural network with high accuracy need to be able to accept both positive and negative inputs. In fact, the ability to accept negative inputs depends on the neuron model. The H-H neuron model is capable of responding to negative inputs; therefore, the hardware H-H neuron circuits based on NbO₂ device can also accept negative inputs, as shown in Fig. R5. For pulse input with short pulse width and a small negative amplitude, it slows down the process of membrane potential integration and hinders the firing of spikes, as shown in Fig. R5a. For persistent input, inputs with values smaller than the negative threshold cannot cause spiking, while values larger than the threshold do, as shown in Figure R5b. The output frequency still increases with the absolute value of the input amplitude. However, in this case, the TS₁ is not switched on, and the spike output of the circuit is caused by the switching on and off of TS₂. Thus, in contrast to the behavior of firing spikes under positive input, there is no process of membrane potential first increasing and then decreasing when the circuit fires spikes. In our work, we did not use negative input but exclusively utilized positive inputs to calculate and demonstrate the firing behavior of neurons, aiming to provide inspiration to the field of H-H neuron applications. However, it's worth noting that we are also exploring the utilization of neuronal behavior in response to negative inputs for computational purposes. Simultaneously, the utilization of H-H neurons in larger networks remains a challenge that requires resolution. This challenge encompasses not only the circuitry of H-H neurons but also the development and application of corresponding algorithms.

Fig. R5 The output of an H-H neuron under negative input. (a) For pulse input with short pulse width and a small negative amplitude, it slows down the process of membrane potential integration and hinders the firing of spikes. (b) For persistent input, inputs with values smaller than the negative threshold do not cause spiking firing, while values larger than the threshold do. The output frequency still increases with the absolute value of the input amplitude.

4. The parameters of the neuron are fixed, defined by the values of the resistances and capacitors. In contrast, biological neural networks typically feature neurons with heterogeneous parameters that can adapt through homeostatic plasticity rules.

Response: We thank the referee for this comment. In biological systems, neurons exhibit plasticity and can adaptively respond to stimuli. Therefore, hardware neurons possessing plasticity are also necessary. However, the mechanisms and applicability of various firing behaviors in H-H neurons are not yet clear. We hope to first establish the relationships between different firing behaviors and the rules for parameter adjustments. Consequently, we initially maintain fixed circuit parameters to investigate whether different firing behaviors can be achieved solely by adjusting the input. Based on this we tried to clarify how to adjust the parameters to obtain other behaviors. With these conditions clarified, we can further enhance the adaptability of H-H neurons by employing memristive and memcapacitive components in place of resistors and capacitors in the circuit. Additionally, by introducing feedback circuits to adjust the resistance of memristive components and the capacitance of memcapacitors, we can construct adaptable and functionally reconfigurable H-H neurons circuits for neuromorphic computation. This will be our long-term goal.

5. Volatile devices, such as TS, used for neurons, are not suitable for implementing synaptic weights, which require non-volatile memories. In the hardware implementation of neural networks, the primary challenge lies in synapses, rather than neurons, as there are orders of magnitude more synapses than neurons. Fabricating

configurable memristors and TS devices requires additional masks, which can significantly increase costs.

Response: We thank the referee for this comment. We agree with the referee's perspective that synapses play a crucial role in hardware implementations of neural networks. However, there is no doubt that the diversity of neuron functions cannot be overlooked. Achieving functional diversity of neurons is of significant importance in the development of neuromorphic technology. Moreover, large-scale synaptic connections between neurons form the foundation for achieving high parallelism in neural network computations. However, for chips applied in engineering, constructing H-H neuron structures using CMOS technology results in complexity and substantial area occupation, severely limiting integration density¹³. As a consequence, in practical applications, CMOS-based H-H neurons require node-level multiplexing, making end-to-end connections unattainable. In the long run, this impacts network parallelism. In contrast, although introducing TS devices during chip fabrication necessitates the additional use of two extra masks, TS-based H-H neuron structures are simpler and exhibit excellent scalability. Thus, they facilitate end-to-end connections between RRAM arrays and neurons, which can fully leverage the high parallelism of RRAM arrays. In this context, we think adding two additional sets of masks cannot be considered a substantial increase in cost.

In addition, a related paper is added in the corresponding locations, and the number of the references is updated accordingly in the revised manuscript.

6. A comparison with CMOS-based analog DPI neurons is essential [1, 2]. It is important to highlight the advantages of the proposed solution. Additionally, a comparison of power consumption would be beneficial.

[1] Arianna Rubino, Melika Payvand, Giacomo Indiveri Ultra-Low Power Silicon Neuron Circuit for Extreme-Edge Neuromorphic Intelligence, IEEE International Conference on Electronics Circuits and Systems (ICECS), 2019

[2] M. V. Nair and G. Indiveri, “An ultra-low power sigma-delta neuron circuit,” in 2019 IEEE International Symposium on Circuits and Systems (ISCAS), May 2019, pp. 1–5.

Response: We thank the referee for this comment. We agree with the reviewer and compare to the CMOS technology based LIF neuron circuit, AdExpIF neuron circuit, and H-H neuron circuit in energy consumption and complexity, as shown in Table 2. Compared to CMOS technology, our neurons are of low complexity and do not require a large number of transistors. At the same time, the energy consumption can be as low as 1.06pJ/spike when the neuron circuit uses the device’s own capacitance instead of the external parallel capacitance. If the device is optimized to further reduce the leakage current and threshold voltage, the energy consumption can be further optimized. Accordingly, we added the table to the supplementary material and the description in the main text on page 19 as follow: “In this case, each spike’s energy consumption of

the circuit can also be reduced to 1.06pJ/Spike (see Supplementary Fig. 20 and Supplementary Table 4).”

Supplementary Table 4. | Comparison between our work and literature reported artificial neuron based on CMOS technology

References	Neuron model	Technology	Energy per spike	Complexity
Besrouer et al ¹⁴	LIF	CMOS 28nm	1.2 fJ	8T+2C
Chen et al ¹⁵	LIF	CMOS 65nm	4 pJ	13T+1C
Rubino et al ¹⁶	AdExp IF	CMOS 22nm	0.99 pJ	56T+4C
Nair et al ¹⁷	AdExp IF	CMOS 180nm	10 pJ	34T+3C
Guo et al ¹⁸	H-H	CMOS 180nm	20 pJ	4T+6R+2C
Ma et al ¹⁹	H-H	CMOS 130nm	170 pJ	43T
Hu et al ²⁰	H-H	Discrete components	-	12Ops+4NPNs+31R+4C
Rutherford et al ²¹	H-H	Discrete components	-	9OPs+21R+9C
This work	H-H	Discrete components	~3 nJ/1.06 pJ *	2TS+2R+2C

*: The energy of our H-H neuron circuit is about 3nJ (@C₁ = 1.55n, C₂ = 330p) and 1.06pJ (@C₁ = 500fF, C₂ = 100fF).

7. Since the input voltage is not a spike but rather a constant voltage, it raises questions about the device's reliability and power consumption. For example, with an input voltage of 0.8V, which corresponds to approximately 80kHz (as shown in Fig. 2f), there is a spike every 12.5 microseconds. Assuming an endurance of 1e10 cycles, the

maximum lifetime would be approximately 35 hours.

Response: We thank the referee for this comment. We agree with the referee that the endurance of the device determines the neuron circuit's lifetime in practical applications. Currently, some articles report that NbO_x can achieve endurance of up to 10¹² cycles²²⁻²⁴. With such endurance, the maximum lifetime of the circuit can reach 3500 hours. Furthermore, neural circuits typically receive pulse inputs rather than continuous inputs in real-world applications, so they are not in constant operation. Therefore, when the operating frequency is reduced, the maximum lifetime of the circuit can be further extended.

8. The impact of intrinsic stochastic switching on TS devices is thoroughly studied and leveraged. However, the effect of conductance device variability from device to device and cycle-to-cycle is not explored.

Response: We thank the referee for this insightful comment. We agree with the referee that the variability of the high and low resistance of the devices can also affect the firing behavior of the circuit output. To explore the influence of the fluctuations of the high and low resistance of the devices, we first performed the statistics of the distribution of the high and low resistance values of the five devices under 1000 cycles. The testing method is show in Fig. R6a, obtaining the distribution of high and low resistance values of the device when the voltage approached V_{TH} . The result is shown in Fig. R6b. When the voltage on the device approaches V_{TH} , the high resistance exhibits fluctuations ranging from 10 K to 20 K, while the low resistance fluctuates within a range of 20 Ω

to 50Ω across different cycles. We take the high and low resistance value fluctuation of device 3 in Figure R6b as a reference, and conduct a simulation to observe the influence of the high and low resistance variation of the two devices on the circuit output when V_{TH} and V_{Hold} fixed. Here, we discuss the two parameters of TS_1 low resistance (R_L) and TS_2 high resistance (R_H) which have the greatest impact on the switching-off time of TS_1 and the switching-on time of TS_2 .

As shown in Fig. R6c, through simulation, we can see that when the low resistance of TS_1 is larger, the on time of TS_1 lasts longer, which is caused by the slower discharge in C_1 . As mentioned in the main text, when the switching-off time of TS_1 is delayed, the circuit will tend to produce bursting firing behavior. However, when the high resistance of TS_2 becomes larger, it will lead to a smaller voltage reduction at V_K in the initial state of the circuit, which means the voltage on TS_2 is closer to V_{TH} of TS_2 . Hence, although the larger high resistance causes the C_2 to charge slower, the lower initial voltage still causes the device TS_2 to turn on earlier. Therefore, when the low resistance of TS_1 is larger and the high resistance of TS_2 is larger, the device is more likely to produce bursting behavior under the same input, otherwise it will produce tonic spiking behavior. Therefore, when the high and low resistance state of the device has a certain randomness, it will also affect the firing behavior of the neuron. To clarify the effect of the high and low resistance variations of the devices on the firing behavior of the output, we also add the following description in the main text: “This occurs because the randomness of V_{TH} , V_{Hold} and the resistance of the two devices affects the relative switching on or off timing of these two TS devices and hence may result in a probability

transition behavior with mixed spiking and bursting rather than an abrupt transition.

First, we explore the effect of the randomness of the V_{TH} and V_{Hold} on the output. Figure.

3a shows 50 DC switching cycles of the two TS devices used under a 400 μA current....

We note that instead of TS₁'s V_{Hold} and TS₂'s V_{TH} , TS₁'s V_{TH} and TS₂'s V_{Hold} also affect the firing features, which we do not discuss here one by one individually. Similarly, we

also explore the effect of the high and low resistance randomness of the device on the circuit output, as shown in Supplementary Fig. 9. It is shown that the combination the

randomness of V_{TH} , V_{Hold} , the high and low resistance values of the two devices results in the probabilistic transition behavior of the circuit output.” We also add the Fig. R6

into the supplementary information.

Fig. R6 The effect of high and low resistance randomness of the devices on the circuit

output. (a) The testing method of the device's high and low resistance values. (b) The distribution of the high and low resistance values of the five devices under 1000 cycles. When the voltage on the device approaches V_{TH} , the high resistance exhibits fluctuations ranging from 10 to 20 K, while the low resistance fluctuates within a range of 20 to 50 Ω across different cycles. (c) The influence of TS_1 low resistance and TS_2 high resistance on the output. when the low resistance of TS_1 is larger, the on time of TS_1 lasts longer, which is caused by the slower discharge in C_1 . As mentioned in the main text, when the switching-off time of TS_1 is delayed, the circuit will tend to produce bursting firing behavior. However, when the high resistance of TS_2 becomes larger, it will lead to a smaller voltage reduction at V_K in the initial state of the circuit, which means the voltage on TS_2 is closer to V_{TH} of TS_2 . Hence, though the larger high resistance will cause the C_2 to charge slower, the lower initial voltage will still cause the device TS_2 to turn on earlier. Therefore, when the low resistance of TS_1 is larger and the high resistance of TS_2 is larger, the device is more likely to produce bursting behavior under the same input, otherwise it will produce tonic spiking behavior.

9. Which circuit is used to convert the LiDAR output into an input for the neuromorphic system? Additionally, what is the latency of a LiDAR system in measuring a distance and converting it into input voltage for the neuromorphic circuit?

Response: Thanks for your detailed questions. For better understanding, we supplied the detailed flowchart of the entire obstacle avoidance approach in Fig.R7. We used CPU to convert the LiDAR output into an input for the neuromorphic system, as shown

in Phase 2 of Fig. R7. And the latency of a LiDAR system in measuring a distance and converting it into input voltage is about 19 ms. In fact, this latency can be divided into two parts. The first part corresponds to the latency of the process represented by Phase 1, which involves the transmission and reception of signals by the lidar, along with preprocessing to obtain distance information. The latency of this phase, namely from Lidar sensing obstacles and sending data packets to industrial computer compressing point cloud data and decoding distance message, is about 19 ms on average of 100 trials. The second part is the latency involves the mapping of distance information to voltage, represented by Phase 2 in Fig. R7, which is about 40 μ s on average of 100 trials. Actually, when conducting the latency comparison between the CMOS implementation and NbO₂ SCNC implementation of our obstacle avoidance approach, we only took the same part, “Phase 3”, in a whole flowchart into consideration. Specifically, we compare the latency from receiving the mapped voltage input to sending movement instructions, because our SCNC plays a role in this phase. At the same time, the latency in Phase1 is actually dependent on the type of tool used to detect distance. When using a lower latency range detector, this part of the latency can be further reduced.

Fig. R7. Detailed flowchart of the entire obstacle avoidance. In Phase 1, Ridar senses and sends data to an industrial computer and then the point cloud data in a packet is compressed and decoded to obtain distance information. In Phase 2, the distance is mapped to a voltage for the convenience of input to a SCNC behavior model. In Phase 3, the SCNC behavioral model supports simulation and output velocity instructions for following motor control.

References:

- 1 Chen, P. *et al.* High-yield and uniform NbOx-based threshold switching devices for neuron applications. *IEEE Trans. Electron Devices* **69**, 2391-2397 (2022).
- 2 Chen, P., Zhang, X., Liu, Q. & Liu, M. NbO₂-based locally active memristors: from physical mechanisms to performance optimization. *Appl. Phys. A* **128**, 1113 (2022).
- 3 Aziz, J., Kim, H., Rehman, S., Khan, M. F. & Kim, D. K. Chemical nature of electrode and the switching response of RF-sputtered NbOx films. *Nanomaterials* **10**(2020).
- 4 Periasamy, P. *et al.* Metal–insulator–metal diodes: role of the insulator layer on the rectification performance. *Advanced Materials* **25**, 1301-1308 (2013).
- 5 Zhao, X. *et al.* Ultrahigh uniformity and stability in NbOx-based selector for 3-D memory by using Ru electrode. *IEEE Transactions on Electron Devices* **68**, 2255-2259 (2021).
- 6 Ding, Y. *et al.* Fatigue of NbOx-based locally active memristors—Part I: experimental characteristics. *IEEE Transactions on Electron Devices*, 1-6 (2023).
- 7 Zhang, X. *et al.* An artificial spiking afferent nerve based on Mott memristors for neurorobotics. *Nat. Commun.* **11**, 51 (2020).
- 8 Nath, S. K. *et al.* Schottky-barrier-induced asymmetry in the negative-differential-resistance response of Nb/NbOx/Pt cross-point devices. *Physical Review Applied* **13**(2020).
- 9 Zhang, X. *et al.* An Artificial Neuron Based on a Threshold Switching Memristor. *IEEE Electron*

- Device Letters* **39**, 308-311 (2018).
- 10 Pickett, M. D., Medeiros-Ribeiro, G. & Williams, R. S. A scalable neuristor built with Mott memristors. *Nat. Mater.* **12**, 114-117 (2013).
- 11 Yi, W. *et al.* Biological plausibility and stochasticity in scalable VO₂ active memristor neurons. *Nat. Commun.* **9**, 4661 (2018).
- 12 Xu, Y., Gao, S., Li, Z., Yang, R. & Miao, X. Adaptive Hodgkin–Huxley neuron for retina-inspired perception. *Adv. Intell. Syst.* **4**, 2200210 (2022).
- 13 Indiveri, G. *et al.* Neuromorphic silicon neuron circuits. *Front Neurosci* **5**, 73 (2011).
- 14 Besrou, M. *et al.* Analog spiking neuron in 28 nm CMOS, in *2022 20th IEEE Interregional NEWCAS Conference (NEWCAS)*, 148-152 (IEEE, 2022).
- 15 Chen, X. *et al.* CMOS-based area-and-power-efficient neuron and synapse circuits for time-domain analog spiking neural networks. *Applied Physics Letters* **122**(2023).
- 16 Rubino, A., Payvand, M. & Indiveri, G. Ultra-low power silicon neuron circuit for extreme-edge neuromorphic intelligence, in *2019 26th IEEE International Conference on Electronics Circuits and Systems (ICECS)*, 19296947, 458-461 (IEEE, 2019).
- 17 Nair, M. V. & Indiveri, G. An ultra-low power sigma-delta neuron circuit, in *2019 IEEE International Symposium on Circuits and Systems (ISCAS)*, 18815532, 1-5 (IEEE, 2019).
- 18 Guo, C., Xiao, Y., Jian, M., Zhao, J. & Sun, B. Design and optimization of a new CMOS high-speed H–H neuron. *Microelectronics J.* **136**, 105774 (2023).
- 19 Ma, Q., Haider, M. R., Shrestha, V. L. & Massoud, Y. Bursting Hodgkin–Huxley model-based ultra-low-power neuromimetic silicon neuron. *Analog Integrated Circuits and Signal Processing* **73**, 329-337 (2012).
- 20 Hu, X. & Liu, C. Dynamic property analysis and circuit implementation of simplified memristive Hodgkin–Huxley neuron model. *Nonlinear Dynamics* **97**, 1721-1733 (2019).
- 21 Rutherford, G. H., Mobbille, Z. D., Brandt-Trainer, J., Follmann, R. & Rosa, E. Analog implementation of a Hodgkin–Huxley model neuron. *American Journal of Physics* **88**, 918-923 (2020).
- 22 Zhang, X. *et al.* Fully memristive SNNs with temporal coding for fast and low-power edge computing, in *2020 IEEE International Electron Devices Meeting (IEDM)*, 20548688, 29.26.21-29.26.24 (IEEE, 2020).
- 23 Ding, Y. *et al.* Forming-free NbOx-based memristor enabling low-energy-consumption artificial spiking afferent nerves. *IEEE Transactions on Electron Devices* **69**, 5391-5394 (2022).
- 24 Luo, Q. *et al.* Nb_{1-x}O₂ based universal selector with ultra-high endurance (>10¹²), high speed (10ns) and excellent V_{th} stability, in *2019 Symposium on VLSI Technology*, 18865282, T236-T237 (IEEE, 2019).

REVIEWER COMMENTS

Reviewer #1 (Remarks to the Author):

The authors have addressed my previous questions in the revision, which might be accepted now.

Reviewer #2 (Remarks to the Author):

Thank you for addressing the previously raised questions; however, I have noticed that my earlier points have only been minimally incorporated into both the main article and supplementary material.

I appreciate the inclusion of experimental and simulation data in the legends of Figures 2 and 3. I would like to emphasize the importance of explicitly stating in the main text, beyond the method section, that the neuron under investigation has not been physically fabricated. Instead, it involves the connection of two Threshold Switching (TS) devices with resistors and capacitors via a printed circuit board (PCB). This clarification ensures a comprehensive understanding of the experimental setup.

Regarding the H-H neuron's response to negative inputs, I understand the observations in Fig. R5 top, where a short pulse width and a small negative amplitude slow down the process of membrane potential integration, impeding the firing of spikes. However, the bottom figure perplexes me, as it seems that the neuron responds similarly to both positive and negative inputs—there's an increase in output frequency with a rise in the absolute value of the input voltage. As per my understanding, negative inputs, typically associated with inhibitory synapses, should result in a reduction of output spiking frequency or possibly give rise to negative spikes.

The authors assert that constructing H-H neuron structures using CMOS technology leads to complexity and significant area occupation, consequently limiting integration density. However, the proposed solution relies on micron-sized devices and necessitates large capacitors. It is crucial to address and quantitatively compare the scalability of the proposed solution with CMOS technologies.

The main text should explicitly address the limitations of the proposed solution and explore potential improvements. The identified weak points include:

1. The fixed parameters of the neuron, determined by the values of resistances and capacitors, prevent the implementation of neuron plasticity.

2. The endurance of the device dictates the neuron circuit's lifetime. The reported value of 10^{12} cycles is from literature for a single device; typically, this value significantly reduces in arrays. Therefore, endurance tests specific to the presented technology should be included, along with an estimated lifetime. While it is acknowledged that neural circuits often receive pulse inputs rather than continuous inputs in real-world applications, the performance of the proposed neurons under pulse inputs, as opposed to continuous voltages, is not clearly discussed. This aspect warrants further exploration and should be addressed in the manuscript.

3. The authors currently utilize a CPU to convert the LiDAR output into an input for the neuromorphic system. In my opinion, the full potential of this bio-inspired approach can only be realized if the sensor is directly linked to the neuromorphic circuit. Otherwise, the obstacle avoidance computation could be carried out on the CPU as well. Could the authors consider a simpler circuit design that directly connects the LiDAR to the neuromorphic circuit?

4. If I understand correctly, the reported 50 times improvement in latency is calculated solely based on the latency from receiving the mapped voltage input to sending movement instructions, which is on the order of hundreds of microseconds. However, it is important to note that the latency of a LiDAR system in measuring a distance and converting it into input voltage is on the order of tens of milliseconds. Therefore, the overall latency improvement appears to be negligible.

We thank reviewer 1 for the recognition of our work and reviewer 2's constructive comments on our manuscript. We revised both the main text and the supplementary information (SI) according to reviewer 2's suggestions.

The changes we made are summarized as follows:

1. Added detailed description to clarify that H-H neurons are built via PCBs.
2. We provided the response of an H-H neuron circuit receiving integration of positive and negative inputs, and added corresponding descriptions in the main text, the results are shown in the added Supplementary Fig. 4.
3. Added a scalability comparison of on-chip memristor-based LIF/H-H circuit with CMOS technology, the results are shown in the added Supplementary Fig. 21. And added discussion about the scalability limitations of the proposed solution and explore potential improvements.
4. Added discussion about the plasticity and endurance limitations of the proposed solution and explored potential improvements.
5. Added discussion about the latency limitations of the LIDAR system.

In the following point-to-point response, the original comments are in black fonts, and our responses are in blue fonts. Changes in the revised main text and SI are highlighted.

One by one Response to Reviewer #2's comments:

Reviewer #2 (Remarks to the Author)

Thank you for addressing the previously raised questions; however, I have noticed that my earlier points have only been minimally incorporated into both the main article and supplementary material.

Response: We thank you for this comment. According to your suggestions, we added more content to the main text and supplementary materials:

- i. Added explanations on page 7 of the main text to provide readers with a clear understanding that our neurons are constructed through a PCB board.
- ii. Added descriptions on page 8 of the main text that our neurons can accept negative inputs and the integration of both positive and negative inputs.
- iii. Added discussion in the Discussion section about the endurance, scalability and plasticity of our neuronal circuit.
- vi. Added discussion in the Discussion section about the latency limitations of the LIDAR system.

1. I appreciate the inclusion of experimental and simulation data in the legends of Figures 2 and 3. I would like to emphasize the importance of explicitly stating in the main text, beyond the method section, that the neuron under investigation has not been physically fabricated. Instead, it involves the connection of two Threshold Switching (TS) devices with resistors and capacitors via a printed circuit board (PCB). This clarification ensures a comprehensive understanding of the experimental setup.

Response: We thank you for this helpful suggestion. We have incorporated explanations on page 7 of the main text to provide readers with a clear understanding that our neurons are constructed through a PCB board. “The H-H neuron circuit comprises two resistors (R_1 , R_2), two capacitors (C_1 , C_2), and two NbO₂-based threshold switching (TS) memristors with d.c.-biased provided by two voltage source (E_1 , E_2). These two TS devices represent the sodium and potassium ion channels in biological neurons, respectively. The device is configured with a Pt/NbO₂/Ti structure (see more fabrication details in Supplementary Fig. 1 and *Methods*). To demonstrate the H-H neuron circuits, we connect the TS devices fabricated in the laboratory with on-the-shelf resistors and capacitors via a printed circuit board (PCB) in this work.”

2. Regarding the H-H neuron's response to negative inputs, I understand the observations in Fig. R5 top, where a short pulse width and a small negative amplitude slow down the process of membrane potential integration, impeding the firing of spikes. However, the bottom figure perplexes me, as it seems that the neuron responds similarly to both positive and negative inputs—there's an increase in output frequency with a rise in the absolute value of the input voltage. As per my understanding, negative inputs, typically associated with inhibitory synapses, should result in a reduction of output spiking frequency or possibly give rise to negative spikes.

Response: We thank you for this comment. In general, the cumulative input of negative pulses does reduce the firing frequency, as you mentioned. However, in addition to this basic input-strength-dependent firing rate behavior, biological neurons also have the

characteristic of inhibition-induced firing, and the firing frequency increases with the increase of the absolute value of input strength, which is one of the 23 special firing behaviors of biological neurons¹. Our neuron circuit is an H-H neuron circuit with high biological confidence, rather than a simple LIF circuit, and thus exhibits an increase in firing frequency as the negative voltage increases. We understand your mention of negative spikes refers to their role in inhibitory synapses and reducing synaptic weights. In biological neurons, spikes are characterized by different spike patterns rather than different polarities¹. To avoid confusion, we modified our manuscript as follows on page 8: “Changing the two capacitors’ capacitances to control the firing features of the output in other literature² follows this principle. In addition, neuron circuits used for neuromorphic computing usually receive both positive and negative input. The H-H neuron circuit we construct is also capable of accepting negative inputs and the integration of both positive and negative inputs (see more details in Supplementary Fig. 4). Based on such an H-H neuron circuit with NbO₂ memristors, we successfully demonstrated 23 natural firing behaviors (see Supplementary Fig. 4 and Supplementary Table 2), as reported in other literatures^{1,3,4}.”

Fig. R1(Fig. S4) Response of an H-H neuron circuit receiving integration of positive and negative inputs. (a) For pulse inputs with short pulse width and small negative amplitude, it slows down the process of membrane potential integration and hinders the firing of spikes. (b) For integration of positive and negative inputs, when the H-H neuron receives a 1.0 V positive input, it fires at a higher frequency. When the H-H neuron receives a -0.7 V voltage at the same time, the firing frequency will be significantly reduced. (c) For constant input, inputs with values smaller than the negative threshold do not cause spiking firing, while values larger than the threshold do. The output frequency still increases with the absolute value of the input amplitude.

3. The authors assert that constructing H-H neuron structures using CMOS technology leads to complexity and significant area occupation, consequently limiting integration density. However, the proposed solution relies on micron-sized devices and necessitates

large capacitors. It is crucial to address and quantitatively compare the scalability of the proposed solution with CMOS technologies.

Response: We thank you for this comment. We agree with you that it is crucial to incorporate a comparison of the scalability between our work and CMOS technology in the manuscript. Although we utilized micrometer-scale devices in this study, the nanometer-scale devices have been demonstrated to be valid by other reported works, and our previous works (NbO_x devices with 60 nm diameter), are also suitable for building such neurons⁵. The capacitors employed in this work were just used to demonstrate the feasibility of proposed neurons and the function of neuronal circuits. In supplementary Fig. 20e, through simulations, we presented that the neuron circuit still works when capacitors are scaled down to the order of pF. A recently published integrated work by Kumar et al. also experimentally demonstrated that the parasitic capacitance of a nanoscale device is valid for building neurons with an integrated resistor⁶. Thus, integrating all devices on chip is possible to build H-H neurons, which is the research content of our following project. We hope your understanding. In such a way, the overall area of the H-H neuron circuit can be estimated to be 4A, where A represents the effective area of RRAM and TS devices. Based on this, we conducted a scalability comparison of the memristor-based LIF circuit, H-H circuit (mLIF / mH-H), with CMOS technology, as shown in Figure R2. The corresponding descriptions have been added to the Discussion section of the main text and supplementary materials:

“...The current H-H neuron circuits are constructed using a PCB board, resulting in a relatively large area that cannot be easily scaled down. A further study of on-chip

integrated H-H neuron circuits is needed to better evaluate the neuron area and performance. A promising scalability comparison of memristor-based neurons and CMOS technology would be conducted, as illustrated in Fig. S22. Only in this way, the advantages of neurons with memristors be decent claimed. In addition, by deploying two SCNCs on the robot, the left or right position of the obstacle relative to the robot can be judged instead of the distance, thus further controlling the turning direction.”

Fig. R2 Comparison the scalability of the memristor-based on-chip LIF / H-H neuron circuits with CMOS technologies⁷⁻¹³.

4. The main text should explicitly address the limitations of the proposed solution and explore potential improvements. The identified weak points include:

Response: We thank you for this comment. We agree with you that it is meaningful to discuss the current limitations of the proposed solution and future optimization

strategies in the manuscript. We provide point-to-point responses and incorporate relevant discussions into the Discussion section of the main text.

(1). The fixed parameters of the neuron, determined by the values of resistances and capacitors, prevent the implementation of neuron plasticity.

Response: At the end of the main text, we discuss the limitations of the proposed solution in terms of neuron circuit plasticity and outline future strategies for enhancing it. “Similarly, increasing the number of SCNCs further improves the resolution of the robot in judging the orientation of the obstacle, which can help the robot achieve more accurate obstacle avoidance behavior. Moreover, the parameters of the current H-H neuron circuit are fixed, which makes the neurons lack plasticity. Further studies on endowing the neurons with plasticity are required, such as replacing the resistors in the circuit with configurable memristors or engineering configurable NbO_x devices. Then such a configurable neuron with learning capability could adjust the memristor’s resistance value after training. So that it could adapt to different dangerous distance ranges with more generality and thus work in various environments, such as indoor and outdoor, slow-motion obstacles, and high-speed obstacles.”

(2). The endurance of the device dictates the neuron circuit’s lifetime. The reported value of 10^{12} cycles is from literature for a single device; typically, this value significantly reduces in arrays. Therefore, endurance tests specific to the presented technology should be included, along with an estimated lifetime. While it is acknowledged that neural circuits often receive pulse inputs rather than continuous

inputs in real-world applications, the performance of the proposed neurons under pulse inputs, as opposed to continuous voltages, is not clearly discussed. This aspect warrants further exploration and should be addressed in the manuscript.

Response: We thank you for this comment. We agree with you that the endurance of the device dictates the neuron circuit's lifetime and the device endurance would be reduced in a large scale in arrays. However, we would like to mention that the optimization of the device in an array level is a huge project on device levels, which is out of the research scope of this work. This work aims to explore the realization of different firing behaviors and novel computing principles based on these behaviors. The circuit's endurance is primarily limited by the endurance of the devices, which falls under fundamental research at the device level and is not the main focus of this work. We agree with you that the neuron circuit often receives pulse inputs, which may slow down the firing frequency of the neurons than constant inputs. However, whether the input is continuous or pulsed, the lifetime is determined by the devices. We believe that there may be some difference on NbO_x devices' endurance, which is unclear and deserves a specific project to study it, thank you. Our previous work⁵ has systematically studied the endurance characteristics among many NbO_x devices, which generally can be enhanced to 10¹³ endurance cycles with a failure-correction method, which means that implementing more than 10¹² endurance cycles among a small number devices is possible. We hope this work could resolve your concerns in a certain.

Anyway, the endurance problem in circuit levels is worthy to be explored for future research but not the focus of this work, which we believe does not affect the novelty of

our work. According to your suggestions, we also added more discussion into the Discussion section of the main text: “On the other hand, the lifetime of H-H neuron circuits is also worth addressing. To improve the lifetime of the neuron circuits, it is more important to improve the endurance of TS devices, which is undoubtedly one of the directions we should continue to pay attention to and study at the device levels. In practical applications, the impact of continuous input, pulse input and device degradation on neurons’ firing behavior is also undoubtedly a topic worthy of further exploration. Nevertheless, our work, as a first step, brings spiking features into intelligent neural circuits, still has important implications, paving the way for real implementation of next-generation high-order brain-like intelligent systems in the future.”

(3). The authors currently utilize a CPU to convert the LiDAR output into an input for the neuromorphic system. In my opinion, the full potential of this bio-inspired approach can only be realized if the sensor is directly linked to the neuromorphic circuit. Otherwise, the obstacle avoidance computation could be carried out on the CPU as well. Could the authors consider a simpler circuit design that directly connects the LiDAR to the neuromorphic circuit?

Response: We thank you for this comment. We agree with you that the full potential of the approach we proposed can be maximized when the sensor is directly connected to the circuit. However, due to limitations imposed by the demonstration platform we utilized, we were unable to extract and process raw information from the laser radar to

directly construct a circuit that converts distance information into voltage. Consequently, we opted to use the CPU of the demonstration platform to perform this function. If in other application scenarios, where resistive sensors such as pressure or photoelectric sensors are directly connected to our neuronal circuit, this intermediary step can be omitted. In this way, it would enable a higher degree of leveraging the advantages of our work, and it also represents an aspect we aim to explore in future research. We added the above discussion to the Discussion section of the main text: “Through adjusting the memristor’s resistance value after training, the neuronal circuit could adapt to different dangerous distance ranges and feature more generality.

Furthermore, the LIDAR system we used requires an additional process to convert the distance signal into a voltage signal through the CPU, which induces an additional delay and hinders our scheme from giving full play to its advantages to some extent. The direct connection between our neuronal circuit to resistive sensors (pressure or photoelectric sensors) can solve the problem and maximize the potential of the proposed solution, which deserves further exploration in the future.”

(4). If I understand correctly, the reported 50 times improvement in latency is calculated solely based on the latency from receiving the mapped voltage input to sending movement instructions, which is on the order of hundreds of microseconds. However, it is important to note that the latency of a LiDAR system in measuring a distance and converting it into input voltage is on the order of tens of milliseconds. Therefore, the overall latency improvement appears to be negligible.

Response: We thank you for this comment. Due to the limitations of the demonstration platform we used, we were unable to reduce the latency of the LiDAR system. Relevant optimizations in this regard fall within the domain of LiDAR research and are not the primary focus of our work. However, your valid concern has prompted us to acknowledge the importance of further exploration in other application scenarios. We will continue investigating other scenarios, in which resistive sensors such as pressure or photoelectric sensors can directly connect to our neuronal circuit, so that we can maximize the potential of the proposed solution. We added the above discussion to the main text: “Through adjusting the memristor’s resistance value after training, the neuronal circuit could adapt to different dangerous distance ranges and feature more generality. Furthermore, the LIDAR system we used requires an additional process to convert the distance signal into a voltage signal through the CPU, which induces an additional delay and hinders our scheme from giving full play to its advantages to some extent. The direct connection between our neuronal circuit to resistive sensors (pressure or photoelectric sensors) can solve the problem and maximize the potential of the proposed solution, which deserves further exploration in the future.”

References:

- 1 Izhikevich, E. M. Which model to use for cortical spiking neurons? *IEEE Trans. Neural. Netw.* **15**, 1063-1070 (2004).
- 2 Bo, Y. *et al.* NbO₂ memristive neurons for burst - based perceptron. *Adv. Intell. Syst.* **2**, 2000066 (2020).
- 3 Yi, W. *et al.* Biological plausibility and stochasticity in scalable VO₂ active memristor neurons. *Nat. Commun.* **9**, 4661 (2018).
- 4 Pickett, M. D., Medeiros-Ribeiro, G. & Williams, R. S. A scalable neuristor built with Mott

- memristors. *Nat. Mater.* **12**, 114-117 (2013).
- 5 Ding, Y. *et al.* Fatigue of NbOx-based locally active memristors—Part I: experimental characteristics. *IEEE T. Electron. Dev.*, 1-6 (2023).
- 6 Kumar, S., Williams, R. S. & Wang, Z. Third-order nanocircuit elements for neuromorphic engineering. *Nature* **585**, 518-523 (2020).
- 7 Guo, C., Xiao, Y., Jian, M., Zhao, J. & Sun, B. Design and optimization of a new CMOS high-speed H–H neuron. *Microelectronics J.* **136**, 105774 (2023).
- 8 Indiveri, G., Chicca, E. & Douglas, R. A VLSI array of low-power spiking neurons and bistable synapses with spike-timing dependent plasticity. *IEEE T. Neural Networ.* **17**, 211-221 (2006).
- 9 Akbari, M., Hussein, S. M., Chou, T.-I. & Tang, K.-T. A 0.3-V conductance-based silicon neuron in 0.18 μm CMOS process. *IEEE T. Circuits-II: Express Briefs* **68**, 3209-3213 (2021).
- 10 Cruz-Albrecht, J. M., Yung, M. W. & Srinivasa, N. Energy-efficient neuron, synapse and STDP integrated circuits. *IEEE T. Biomed. Circ. S.* **6**, 246-256 (2012).
- 11 Joubert, A., Belhadj, B., Temam, O. & H'eliot, R. Hardware spiking neurons design: analog or digital?, in *the 2012 International Joint Conference on Neural Networks (IJCNN)*, 1-5 (IEEE, 2012).
- 12 Moradi, S., Bhave, S. A. & Manohar, R. Energy-efficient hybrid CMOS-NEMS LIF neuron circuit in 28nm CMOS process, in *the 2017 IEEE Symposium Series on Computational Intelligence (SSCI)*, 1-5 (IEEE, 2017).
- 13 Besrour, M. *et al.* Analog spiking neuron in 28 nm CMOS, in *the 2022 20th IEEE Interregional NEWCAS Conference (NEWCAS)*, 148-152 (IEEE, 2022).

REVIEWERS' COMMENTS

Reviewer #2 (Remarks to the Author):

The authors addressed the limitations and potential improvements of the proposed solution, as requested in my comments.